

# Continuous decline in lower stratospheric ozone offsets ozone layer recovery

William T. Ball[1,2], Justin Alsing[3], Daniel J. Mortlock[4,5,6], Johannes Staehelin[2],
Joanna D. Haigh[4,7], Thomas Peter[2], Fiona Tummon[2], Rene Stübi[8],
Andrea Stenke[2], John Anderson[9], Adam Bourassa[10], Sean M. Davis[11,12],
Doug Degenstein[10], Stacey Frith[13,14], Lucien Froidevaux[15], Chris Roth[10],
Viktoria Sofieva[16], Ray Wang[17], Jeannette Wild[18,19], Pengfei Yu[11,12],
Jerald R. Ziemke[14,20], and Eugene V. Rozanov[1,2]

[1]Physikalisch-Meteorologisches Observatorium Davos World Radiation Centre, Dorfstrasse 33, 7260 Davos Dorf, Switzerland
[2]Institute for Atmospheric and Climate Science, Swiss Federal Institute of Technology Zurich, Universitaetstrasse 16, CHN, CH-8092 Zurich, Switzerland
[3]Center for Computational Astrophysics, Flatiron Institute, 162 5th Ave, New York, NY 10010, USA
[4]Physics Department, Blackett Laboratory, Imperial College London, SW7 2AZ, UK
[5]Department of Mathematics, Imperial College London, SW7 2AZ, UK
[6]Department of Astronomy, Stockholms universitet, SE-106 91 Stockholm, Sweden
[7]Grantham Institute - Climate Change and the Environment, Imperial College London, SW7 2AZ, UK
[8]Federal Office of Meteorology and Climatology, MeteoSwiss, CH-1530 Payerne, Switzerland
[9]Hampton University, Hampton, VA, USA
[10]Institute of Space and Atmospheric Studies, University of Saskatchewan, Saskatoon, Canada
[11]Cooperative Institute for Research in Environmental Sciences, University of Colorado, Boulder, CO, USA
[12]NOAA Earth System Research Laboratory, Boulder, CO, USA
[13]NASA Goddard Space Flight Center, Silver Spring, MD, USA
[14]Science Systems and Applications Inc., Lanham, MD, USA
[15]Jet Propulsion Laboratory, California Institute of Technology, Pasadena, CA, USA
[16]Finnish Meteorological Institute, Helsinki, Finland
[17]School of Earth and Atmospheric Sciences, Georgia Institute of Technology, Atlanta, GA, USA
[18]NOAA/NWS/NCEP/Climate Prediction Center, College Park, MD, USA
[19]Innovim LLC, Greenbelt, MD, USA
[20]NASA Goddard Space Flight Center, Greenbelt, MD, USA

*Correspondence to:* W. T. Ball (william.ball@env.ethz.ch)



**Abstract.** Ozone forms in the Earth's atmosphere from the photodissociation of molecular oxygen, primarily in the tropical stratosphere. It is then transported to the extratropics by the Brewer-Dobson circulation (BDC), forming a protective 'ozone layer' around the globe. Human emissions of halogen-containing ozone-depleting substances (hODSs) led to a decline in stratospheric ozone until

they were banned by the Montreal Protocol (MP), and since 1998 ozone in the upper stratosphere shows a likely recovery. Total column ozone (TCO) measurements of ozone between the Earth's surface and the top of the atmosphere, indicate that the ozone layer has stopped declining across the globe, but no clear increase has been observed at latitudes outside the polar regions ($60°–90°$). Here we report evidence from multiple satellite measurements that ozone in the lower stratosphere

between $60°$S and $60°$N has declined continuously since 1985. We find that, even though upper stratospheric ozone is recovering in response to the MP, the lower stratospheric changes more than compensate for this, resulting in the conclusion that, globally ($60°$S–$60°$N), stratospheric column ozone (StCO) continues to deplete. We find that globally, TCO appears to not have decreased because tropospheric column ozone (TrCO) increases, likely the result of human activity and harmful

to respiratory health, are compensating for the stratospheric decreases. The reason for the continued reduction of lower stratospheric ozone is not clear, models do not reproduce these trends, and so the causes now urgently need to be established. Reductions in lower stratospheric ozone trends may partly lead to a small reduction in the warming of the climate, but a reduced ozone layer may also permit an increase in harmful ultra-violet (UV) radiation at the surface and would impact human and

ecosystem health.

## 1 Introduction

The stratospheric ozone layer protects surface life from harmful solar ultraviolet radiation. In the second half of the 20th century, halogen-containing ozone depleting substances (hODSs) resulting from human activity, mainly in the form of chloroflurocarbons (CFCs), led to the decline of the

ozone layer (Molina and Rowland, 1974). The clearest example of ozone depletion was signified by the formation of an ozone hole over the Southern polar region, but even outside there was a clear reduction in total coulmn ozone (TCO) (Farman et al., 1985; WMO/NASA, 1988; WMO, 2011, 2014). The Montreal Protocol came into effect in 1989, banning multiple substances responsible for ozone layer depletion, and by 1997 it became apparent that a decline in TCO had ceased at almost

all non-polar latitudes.

    The general expectation is that global mean stratospheric column ozone (StCO) will increase as hODSs continue to decline, but an attribution of increasing TCO to decreasing ODSs has not yet been possible (WMO, 2014); a cooling stratosphere is also thought to aid the recovery of ozone by slowing temperature-dependent reaction rates. Models predict that mean TCO will increase, but



this also remains uncertain since projections rely substantially on the $CO_2$, $N_2O$ and $CH_4$ emissions scenarios.

Only recently has a TCO recovery been detected during the austral spring (Solomon et al., 2016). However, elsewhere observations of global TCO levels have remained stable since 2000 (WMO, 2014), with most latitudes displaying a positive, but non-significant, decadal trend (WMO, 2014).

Results from Frith et al. (2014) suggest a potential peak in positive trends around 2011, after which positive trends declined while uncertainties increased, despite longer timeseries.

In the past the attribution and identification of ozone recovery was made through multiple linear regression (MLR) analysis with most studies considering either piecewise linear trends (PWLT) to represent trends, with an inflection date usually at the end of 1997, or the equivalent effective

stratospheric chlorine (EESC) proxy to represent the influence of hODSs on ozone, which is a non-linear smoothly-varying proxy with an inflection date also around 1997 (Newman et al., 2007). Two recent studies, Chehade et al. (2014) and Frith et al. (2014), both investigated changes in TCO observations up to 2012 and 2013, respectively, and came to similar conclusions: trends using PWLT or EESC prior to 1997 agree that ozone declined to a minimum in 1997, but from 1997 the use of the

EESC proxy suggests a significant and positive increase at all latitudes (larger at higher latitudes), while use of PWLT shows peaks at mid-latitudes but is generally lower and non-significant at most latitudes outside the polar regions. These results suggest that attribution to EESC prior to 1997 is the dominant contributor to the long-term trend, but after 1997 it may be less representative, or that large dynamical variability is interfering with post-1997 trends. As both Chehade et al. (2014) and

Frith et al. (2014) note, the post-1997 EESC estimate is partially locked by the large decline during the pre-1998 period, while in PWLT the period to fix the estimate is not influenced by the latter or earlier period, respectively. The consequence may be, then, that a significant post-1997 change in TCO might indeed represent a hODS-related increase in ozone, but this may be embedded within ozone that is not actually increasing, or increasing at a slower rate, as shown by the PWLT that

represents the overall timeseries without any specific physical attribution.

Despite a lack of clear recovery in TCO, ozone in the upper stratosphere above 10 hPa appears to be recovering, which has been reported with significant positive decadal trends in vertical profiles, and altitude-latitude spatial maps, from multiple ozone composites that merge observations from various space missions, especially at mid-latitudes (Kyrölä et al., 2013; Laine et al., 2014; WMO,

2014; Tummon et al., 2015; Harris et al., 2015; Steinbrecht et al., 2017; Ball et al., 2017; Sofieva et al., 2017; Bourassa et al., 2017). Trends in these results are almost always presented as percentage-change per decade, which does not illuminate the contribution to the column ozone changes. Thus, noting that total column may be increasing, and that upper stratospheric ozone is recovering, does not mean that stratospheric ozone as a whole is increasing as counteracting trends in different layers of

the stratosphere, and troposphere, may counteract a TCO recovery. Indeed, if TCO does not display any significant changes since 1997, while the upper stratosphere displays significant increases, then



either the uncertainties due to unattributed dynamical variability interfere in the significance of the trend determined through MLR, or there are counteracting trends at lower levels of the stratosphere, or in the troposphere.

Suggestions of a decrease in lower stratospheric ozone have been presented elsewhere (Kyrölä et al., 2013; Gebhardt et al., 2014; Sioris et al., 2014; Nair et al., 2015; Vigouroux et al., 2015). However, it has been difficult to confirm (WMO, 2014) because: (i) ozone is typically integrated over wide latitude bands and/or total column ozone (TCO) is considered, both of which may lead to cancellation of opposing trends; (ii) large dynamical variability unaccounted for in regression
analysis together with shorter timeseries lead to higher uncertainties (Tegtmeier et al., 2013); (iii) below 20 km there are large ozone gradients, with low ozone close to the tropopause; and (iv) composite-data merging techniques have hindered identification of robust changes (Harris et al., 2015; Ball et al., 2017). Despite all of these issue, uncertainties between limb sounding instruments have been reported to be less than ~10–15% near 16 km (Tegtmeier et al., 2013).

In addition to only reporting decadal percentage changes, most studies typically do not consider altitudes below 20 km (~60 hPa), missing stratospheric changes down to 16 km in the tropics (30°S–30°N) or ~12 km at mid-latitudes (60°–30°), regions that contain a large fraction of, and drive most sub-decadal variability in, TCO. A recent study by Bourassa et al. (2017) extended their analysis of the SAGE-II/OSIRIS ozone composite down to 18 km, where widespread, partially significant,
negative ozone trends (1998–2016, PWLT) can be seen at all latitudes from 50°S to 50°N. Models do predict a decline in lower stratospheric ozone in the tropics in the future (e.g. by 2060; Eyring et al. (2010); WMO (2011)), but evidence for Brewer-Dobson circulation (BDC) driven decreases up to present day continue to remain weak because multiple observations since 2000 do not support such a tropical stratospheric decline, which has levelled off since 2000, and decreases that have been
identified between 32–36 km (near 10 hPa) km are thought to be largely due to the 2000–2003 period of high ozone levels WMO (2014), and so may currently be an artefact of the analysis period rather than a change in the BDC.

Finally, issues remain in ozone timeseries analysis from both the use of the standard analysis technique employing ordinary least squares, multiple linear regression (MLR) that can lead to biased
estimates (Ball et al., 2017) due to unaccounted-for residual variance, time of day and geolocation biases (Sofieva et al., 2014), vertical and horizontal spatial resolution (Kramarova et al., 2013), and the presence of satellite drifts and biases introduced into composite timeseries from how they were merged (Tummon et al., 2015; Harris et al., 2015; Ball et al., 2017). All of these can lead to conflicting results between composite datasets, even those constructed using similar underlying data
sources (WMO, 2014; Ball et al., 2017).

Our aim here is to quantify the absolute changes in ozone and contribution to TCO since 1998, i.e. not simply their relative change in percentage. We bring to bear a robust regression analysis approach (section 2.1) through dynamical linear modelling (DLM) (Laine et al., 2014; Ball et al.,



2017). The major step forward that DLM provides is the estimation of smoothly varying, non-linear
background trends, without the need to provide a prescribed EESC explanatory variable. Although
this precludes a clear physical attribution, similar to PWLT, it allows for an assessment of how
ozone is evolving on decadal and longer timescales, e.g. to identify if and when an inflection in
ozone occurs. We also make use of updated ozone composites (section 3), extended to 2015/6, and
we begin by considering relative percentage changes since 1998 to put these new data, analysed with
the DLM approach, in the context of previously reported trends for relative changes above 20 km,
but extended down to the tropopause (section 4.1). From there, we consider the absolute contribution
of partial column ozone (PCO) from the whole stratosphere (StCO), the upper (1–10 hPa, ~32–48
km), middle (10–32 hPa, ~25–32 km), and lower stratosphere (32–147 hPa or ~13–25 km at mid-
latitudes; 32–100 hPa or ~16–25 km in subtropical and equatorial latitudes; section 4.2), and then
the tropospheric contribution (section 4.3). We finally consider what two chemistry climate models
in specified dynamics mode suggest trends have been (section 4.4). We discuss our findings and
conclude in section 5.

## 2 Methods

### 2.1 Regression analysis

The standard method to estimate decadal trends or changes in ozone, multiple linear regression
(MLR), is known to have estimator bias and regressor aliasing (Marsh and Garcia, 2007; Chiodo
et al., 2014). To minimise these effects we use a more robust method using a Bayesian inference
approach through Dynamical Linear Modelling (DLM) (Laine et al., 2014; Ball et al., 2017). DLM
(Laine et al., 2014) is similar to MLR in that the same regressors (see section 2.2, below) are used
for known drivers of ozone variability, and an autoregressive term is included. However, the trend
is not predetermined with a linear, or piece-wise linear, model, but is allowed to slowly vary in
time, and the degree of trend non-linearity is an additional free parameter to be jointly inferred from
the data. We infer posterior distributions on the non-linear trends by Markov Chain Monte Carlo
(MCMC) sampling. DLM analyses typically have more conservative uncertainties than MLR since
they are based on a more flexible model, and formally integrate over uncertainties in the regression
coefficients, seasonal cycle and dynamics, autoregressive coefficients and parameters characterizing
the degree of non-linearity in the trend. The time-varying, background changes are estimated, rather
than specified by, for example, an estimate of equivalent effective stratospheric chlorine (EESC)
(Newman et al., 2007) or a linear-trend; there is no need for assumptions about when and where a
decline in hODSs occurs.




## 2.2 Regressor variables

Similar to MLR, we use regressor timeseries that represent known drivers of stratospheric ozone variability. These include: the F30 cm radio flux as a solar proxy (as it better represents UV variability than the commonly used F10.7 cm flux (Dudok de Wit et al., 2014)), a latitudinally resolved stratospheric aerosol optical depth (SAOD) for volcanic eruptions (Thomason et al., 2017), an ENSO index (NCAR, 2013) representing El niño Southern Oscillation variability[1], the Quasi-Biennial Oscillation at 30 and 50 hPa[2]. We use the Arctic and Antarctic Oscillation[3] proxy for Northern and Southern TCO and partial column ozone (PCO) trend estimates. We use a second order autoregressive (AR2) process (Tiao et al., 1990) to avoid the auto-correlation of residuals. We remove the two year period June 1991 to May 1993, inclusive, from the analysis to avoid problems related to impacts of satellite ozone retrieval due to stratospheric aerosol loading (Davis et al., 2016), and aliasing between regressors within the regression analysis (Chiodo et al., 2014); the volcanic aerosols still show slowly varying changes, which are important to consider as a regressor since this has a larger impact on ozone in the lower stratosphere than the upper.

## 2.3 Statistics

We do not apply any statistical tests, which therefore avoids making assumptions about the (posterior) distributions being considered. The posteriors presented in all figures represent the full information about the change in ozone since 1998 obtained from the DLM analysis; these are not always normally distributed. The probabilities discussed, and presented represent the percentage of the total number of DLM samples (n=100,000) that decrease in ozone; positive increases have values less than 50% and therefore increases at 80, 90 and 95% probabilities are indicated by their respective contours in Fig. 1 and A1, and have values less than or equal to 20, 10 and 5% in Figs. 2, A3, A4, A6, A9, and A10.

## 3 Ozone Data

### 3.0.1 Satellite ozone composites

A summary of the ozone merged datasets – SWOOSH (Davis et al., 2016), GOZCARDS (Froidevaux et al., 2015), SBUV-MOD (Frith et al., 2014), SBUV-Merged-Cohesive (Wild and Long, 2017), SAGE-II/CCI/OMPS (Sofieva et al., 2017) and SAGE-II/OSIRIS/OMPS (Bourassa et al., 2014) – and an intercomparison of the publicly available data up to 2012 can be found in Tummon et al. (2015); data up to 2016 are available upon request from respect composite PIs (see also Steinbrecht et al., 2017). These data are monthly, zonally averaged, homogenised, and bias-corrected ozone

---

[1]From NOAA: http://www.esrl.noaa.gov/psd/enso/mei/table.html
[2]From Freie Universitaet Berlin: http://www.geo.fu-berlin.de/en/met/ag/strat/produkte/qbo/index.html.
[3]From http://www.cpc.ncep.noaa.gov/products/precip/CWlink/daily_ao_index/teleconnections.shtml.





datasets. We consider the period 1985–2016 in all cases, except SAGE-II/CCI/OMPS up to 2015, as it ends in July 2016. We consider the latitudinal range 60°S to 60°N where all datasets have latitudinal coverage, and from 13 to 48 km in SAGE-II/CCI/OMPS and SAGE-II/OSIRIS/OMPS, the

175 approximately equivalent pressure range of 147–1 hPa that we consider in SWOOSH, GOZCARDS, and Merged-SWOOSH/GOZCARDS, and 50–1 hPa in SBUV-NOAA, SBUV-NASA, and Merged-SBUV.

### 3.0.2 Merged-SWOOSH/GOZCARDS and Merged-SBUV

SWOOSH and GOZCARDS are composites constructed with similar instrument data (Tummon et al., 2015), but with different pre-processing and merging techniques; the same is true for SBUV-MOD and SBUV-Merged-Cohesive, which are constructed using nadir-viewing backscatter instruments. The Merged-SWOOSH/GOZCARDS and Merged-SBUV results presented here combine these two pairs of composites, which show slightly different spatial variability (Fig. A1) (Tummon

et al., 2015; Harris et al., 2015; Steinbrecht et al., 2017). Part of the reason is related to offsets and drifts in the data that continue to be one of the largest remaining sources of uncertainty within, and between, ozone composites (Harris et al., 2015; Ball et al., 2017). These artefacts can be largely accounted for using the methodology developed by (Ball et al., 2017), which we apply to both pairs of data separately; examples of corrected timeseries in the lower stratosphere are given in Fig. A2,

and others can be found in Ball et al. (2017). This method also fills data gaps, which is reasonable if they are discontinuous for only a few months. This is true for these datasets, but is not for the SAGE-II/CCI/OMPS and SAGE-II/OSIRIS/OMPS. SWOOSH, SBUV-Merged-Cohesive and GOZCARDS have been updated since previous intercomparisons (Tummon et al., 2015; Harris et al., 2015). GOZCARDS v2.20, used here, includes SAGE-II v7.0 and has a finer vertical resolution than

earlier versions. It must be stressed that the resolution of SBUV-instruments below 22 hPa (25 km) is low (McPeters et al., 2013; Kramarova et al., 2013), so linear trends estimated at 25–46 hPa also encompass altitudes lower than those that they formally represent (see section 4 for a discussion on this).

### 200 3.0.3 Total column ozone

We use merged SBUV v8.6 (Frith et al., 2014) for comparison of results with total column ozone (TCO) observations, which are available on a 5° latitude grid from 1970 onwards. We verify stability of SBUV TCO after 1997 by comparing SBUV TCO overpass data with the independent Arosa ground measurements, which are available from 1926 to present (Scarnato et al., 2010).



### 3.0.4 Tropospheric column ozone

For tropospheric ozone, we consider OMI/MLS tropospheric column ozone measurements, discussed by Ziemke et al. (2006). The tropospheric ozone are estimated through a residual method that derives daily maps of tropospheric column ozone (TrCO) by subtracting MLS stratospheric column ozone (StCO) from co-located OMI total column ozone. The OMI/MLS data, including data quality and data description, are publicly available[4]. Coverage of the OMI/MLS ozone is monthly (October 2004–present) and at $1° \times 1.25°$ horizontal resolution, which we have zonally averaged to make comparisons here.

## 4 Results

### 4.1 Latitude-altitude resolved post-1997 ozone changes

Concentrations of active stratospheric hODSs reached a maximum in ~1997 (Newman et al., 2007), and vertically-resolved satellite measurements show evidence that upper stratospheric ozone (10–1 hPa; ~32–48 km) started recovering soon after (WMO, 2014). Fig. 1 presents post-1998 ozone changes from four ozone composites that combine multiple satellite instruments (see section 3). The Merged-SBUV and Merged-SWOOSH/GOZCARDS composites show 95% probability that upper-stratospheric ozone at all latitudes between 60°S and 60°N has increased. This is less robust in SAGE-II/CCI/OMPS and SAGE-II/OSIRIS/OMPS, which show differences at equatorial latitudes (10°S–10°N). The reason for the difference is not clear, but we note that in this region nearly 50% of the data are missing in the first five years (1998–2002), while Merged-SWOOSH/GOZCARDS and Merged-SBUV have no missing data (Harris et al., 2015).

In contrast to the upper stratosphere, all four composites show a consistent ozone decrease below 32 hPa / 24 km at all latitudes (Fig. 1). The regions where probabilities are high (>80, 90 and 95%, see legend) are similar in all composites, except for Merged-SBUV which has a lower vertical resolution. Right of Fig. 1a are two examples of the Merged-SBUV vertical resolution, indicating the contribution to ozone at a particular layer, at tropical (solid) and Northern mid-latitudes (dashed) (Kramarova et al., 2013). The profiles peaking at 3 hPa (red) span ~1–8 hPa, and contain only upper stratospheric changes. However, while changes at 25 hPa (blue) show insignificant changes in the other higher resolution composites, the Merged-SBUV profile ranges ~15–100 hPa, thus including the lowest part of the stratosphere where changes in the other composites are negative. We cannot use Merged-SBUV for comparison of resolved ozone changes, although a TCO product based upon these data can be used for comparison later (section 4.3). While Merged-SBUV has a different spatial pattern, the increases in the upper, and decreases in the lower, stratosphere qualitatively agree with

---

[4]From the NASA Goddard website https://acd-ext.gsfc.nasa.gov/Data_services/cloud_slice/





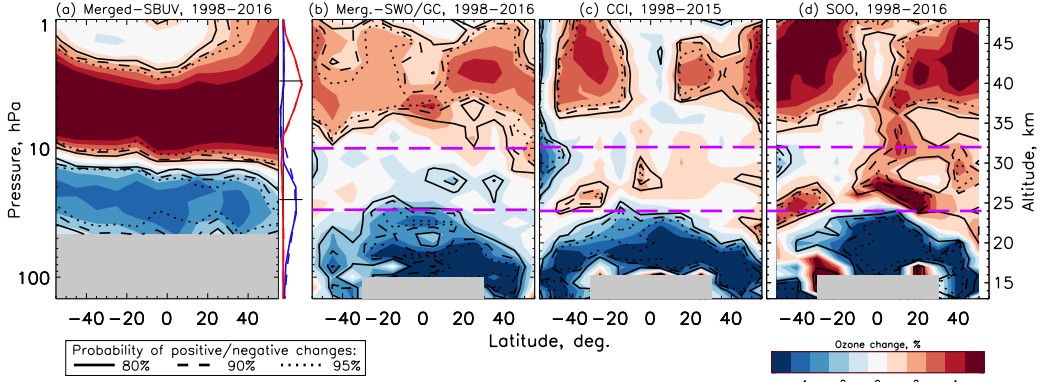

Figure 1: Zonally averaged change in ozone between 1998 and 2016. From (left to right) the Merged-SBUV, Merged-SWOOSH/GOZCARDS, SAGE-II/CCI/OMPS (CCI), and SAGE-II/OSIRIS/OMPS (SOO) composites (see methods for abbreviations and ozone-change calculations). Red represents increases, blue decreases (%; see legend). Contours represent probability levels of positive or negative changes. Grey regions represent unavailable data. Pink dashed-lines delimit regions integrated to partial ozone columns in Figs. 2, A3, A4, A6, A9, and A10. To the right of Merged-SBUV are the instrument observing profiles centred at 3 hPa (red, upper) and 25 hPa (blue) at Northern mid-latitudes (dashed) and in the tropics (solid), from Kramarova et al. (2013). SAGE-II/CCI/OMPS changes are for 1998–2015.

the other composites. These results clearly indicate that ozone has declined in the lower stratosphere
since 1998.

We note that our spatial results (Figs. 1 and A1) show similar patterns and changes to those presented in other studies, (e.g.WMO (2014); Bourassa et al. (2014); Sofieva et al. (2017); Steinbrecht et al. (2017)), though these typically do not extend below 20 km and so often do not show the extensive decrease in lower stratospheric ozone that we do. Bourassa et al. (2017) extend down to 18
km and, indeed, show a larger region of decreasing ozone trends, but even this does not extend as far down as our results, i.e. ~17 km for 30°S–30°N, and 13 km outside this region. Our results do not qualitatively disagree with previous studies and approaches (WMO, 2014). However, four additional years of data (Tummon et al., 2015; Harris et al., 2015), an improved regression analysis method (Laine et al., 2014; Ball et al., 2017) (see section 2), and techniques to account for data artefacts
(Ball et al., 2017), means we are now able to confidently identify changes in the lower stratosphere.

### 4.2    Stratospheric and Total Column Ozone post-1997 changes

The spatial trends presented in Fig. 1 are informative for understanding where, and assessing why, changes in stratospheric ozone are occurring. However, stratospheric ozone changes are usually reported as decadal percentage change vertical profiles or spatial maps (e.g. as in Fig. 1), which



hides the absolute changes in ozone, and the contribution to the total column, which are almost never
reported. A recovery in the upper stratosphere is important to identify, but this region contributes a
smaller fraction to the total column than the middle and lower stratosphere. Thus, smaller percentage
changes over a reduced altitude range in the lower stratosphere can actually produce larger integrated
changes than in the more extended regions higher up.

In Fig. 2 we present changes in partial column ozone (PCO) in Dobson Units (DU) from Merged-
SWOOSH/GOZCARDS for the whole stratospheric column (StCO), and for the upper (10–1 hPa),
and lower stratosphere (147–32 hPa or 13–24 km at >30°; 100–32 hPa or 17–24 km at <30°), respec-
tively. We note that the tropopause, the boundary layer between the troposphere and stratosphere,
varies seasonally, but is on average around 16 km (tropics) and 10–12 km (mid-latitudes); our con-
servative choice of slightly higher altitudes ensures that we avoid including the troposphere. Due to
the near-complete temporal and vertical coverage, we focus on the Merged-SWOOSH/GOZCARDS
composite (SAGE-II/OSIRIS/OMPS and SAGE-II/CCI/OMPS are provided in Figs. A3 and A4,
respectively[5]). Fig. 2 shows posterior distributions of the 1998–2016 ozone changes, with black
numbers representing the percentage of the distribution that is negative, in 10° bands (left) and in-
tegrated 'global' (defined as 60°S–60°N) PCO (right), along with the TCO observed by SBUV (red
curves and numbers; upper row).

Upper stratospheric ozone (Fig. 2, middle row) has increased since 1998 in almost all latitude
bands, in half the cases at >90% probability, and >95% at 40°–60° in both hemispheres. Globally,
the probability exceeds 99% that upper stratospheric ozone has increased, confirming that the MP
has indeed been successful in reversing trends in this altitude range.

Changes in the lower stratosphere (Fig. 2, lower row) show ozone decreases, typically exceeding
90% probability (50°S–50°N). There is 99% probability that lower stratospheric ozone has decreased
globally (60°S–60°N) since 1998; SAGE-II/OSIRIS/OMPS and SAGE-II/CCI/OMPS both support
this result with 87 and 99% probabilities, respectively (Figs. A3 and A4).

Integrating the whole stratosphere vertically, to form the stratospheric column ozone (StCO;
Fig. 2, upper row), we see that all distributions imply a decrease (i.e. values >50%); probability is
generally higher in tropical latitudes (30°S–30°N). Integrating over all latitudes, global StCO (right)
indicates that stratospheric ozone has decreased with >90% probability. We compare the Merged-
SWOOSH/GOZCARDS change with SBUV TCO, the latter of which includes both troposphere and
stratosphere. The global SBUV TCO indicates that ozone has, in contrast to the StCO, changed little
compared to 1998.

---

[5]It should be noted that while each latitude band PCO of SAGE-II/OSIRIS/OMPS and SAGE-II/CCI/OMPS typically has
between 60 and 90% of months where data are available for 1985–2015/6, integrating bands across all latitudes leads to a re-
duction of available months (see Fig. A5), though estimates of the change since 1998 can still be made and uncertainties due to
the reduced data are captured in the posteriors given in Figs. A3 and A4; this does not affect Merged-SWOOSH/GOZCARDS
or SBUV TCO.





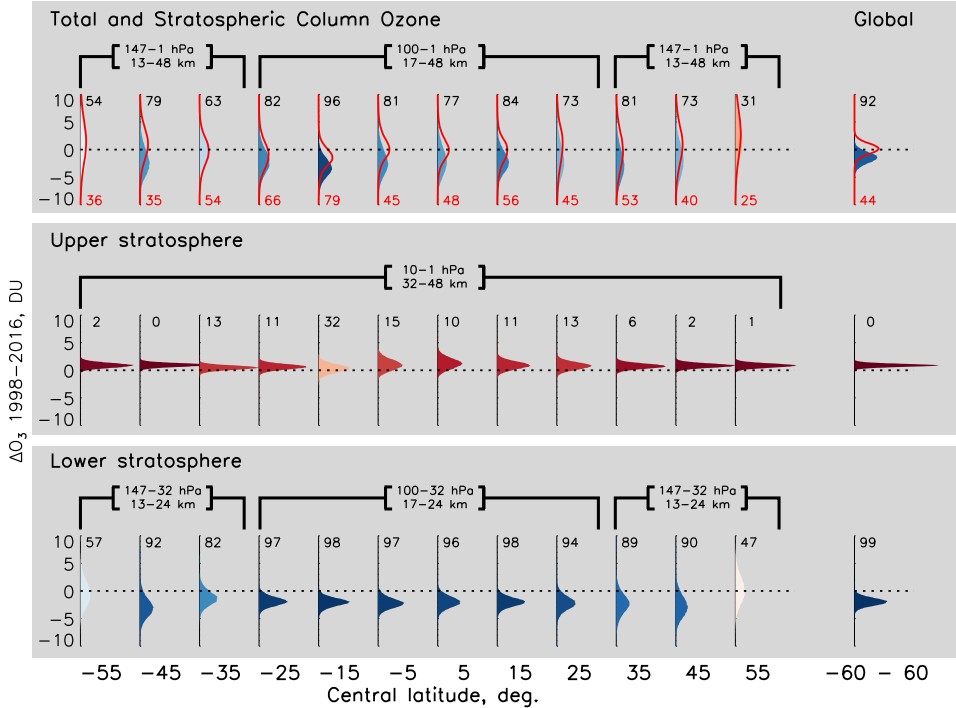

Figure 2: Merged-SWOOSH/GOZCARDS posterior distributions (shaded) for the 1998–2016 total and partial column ozone changes. (Top) whole stratospheric column, (middle) upper and (bottom) lower stratosphere in 10° bands for all latitudes (left) and integrated from 60°S–60°N ('Global', right). The stratosphere extends deeper at mid-latitudes than equatorial (marked above each latitude). Numbers above each distribution represents the distribution-percentage that is negative; colours are graded relative to the percentage-distribution (positive, red-hues, with values <50; negative, blue). SBUV total column ozone (red curves) is given in the upper row and negative distribution-percentages are given as red numbers.

We note that uncertainty remains in the middle stratosphere (Fig. A6), with Merged-SWOOSH/GOZCARDS, SAGE-II/CCI/OMPS, and SAGE-II/OSIRIS/OMPS displaying different changes. SAGE-II/OSIRIS/OMPS, in particular, shows a significant positive trend, which leads to the global StCO indicating no change since 1998 (Fig. A3). This is likely a result of how the data were merged to form composites (see example in Fig A7 at northern mid-latitudes and 30 km), and is an issue that remains to be resolved (Harris et al., 2015; Ball et al., 2017; Steinbrecht et al., 2017). Nevertheless, the changes in the upper and lower stratosphere are consistent in all ozone composites, and a globally-integrated StCO decline is indicated by both Merged-SWOOSH/GOZCARDS-$O_3$ and SAGE-II/CCI/OMPS-$O_3$.



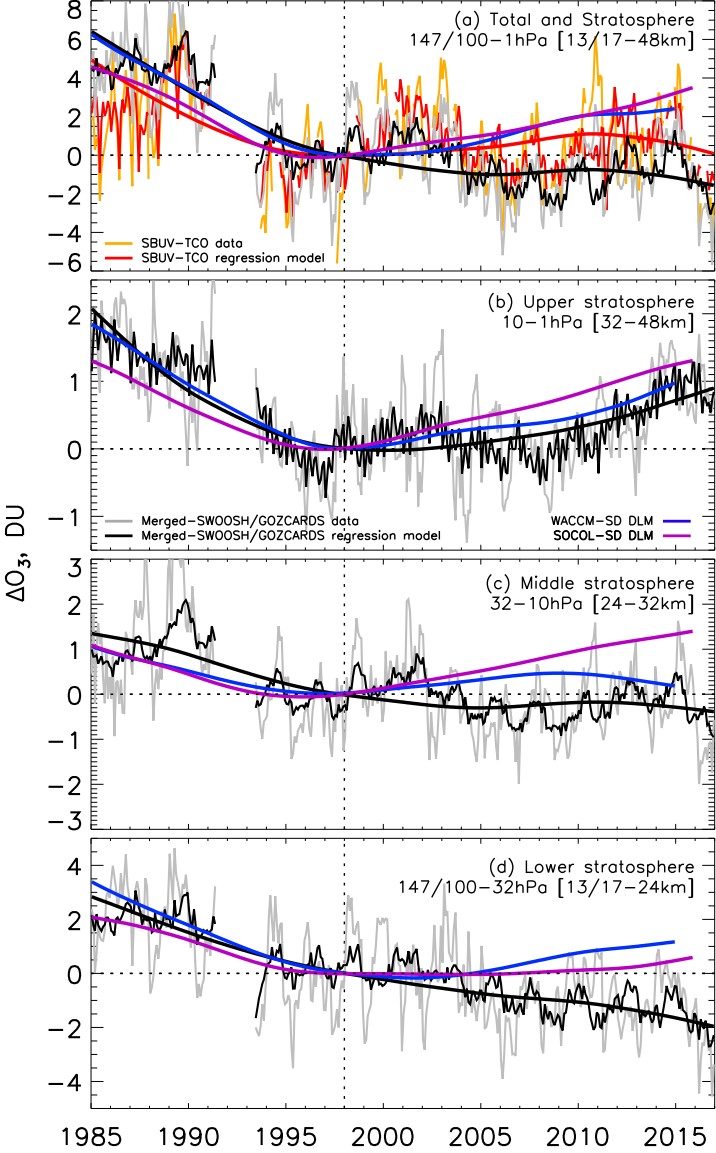

Figure 3: 'Global' 60°S–60°N 1985–2016 total (TCO) and partial (PCO) column ozone anomalies. Deseasonalised and regression model timeseries are given for the Merged-SWOOSH/GOZCARDS merged composite (grey and black, respectively) for (a) the whole stratospheric column, (b) upper, (c) middle, and (d) lower stratospheric PCOs. The DLM non-linear trend is the smoothly varying thick black line. In (a), the deseasonalised SBUV TCO is also given (orange), with the regression model (red) and the non-linear trend (thick, red). Data are shifted so the trend-line is zero in 1998. DLM results for WACCM-SD (blue) and SOCOL-SD (purple) from Fig. A11 are also shown.



To make these globally-integrated results clear, we show in Fig. 3a the SBUV TCO (yellow/red)
and Merged-SWOOSH/GOZCARDS StCO (grey/black); in all of the panels in Fig. 3, the timeseries
are bias-shifted so that the smoothly varying non-linear trend crosses the zero line in January 1998,
so that relative changes can be clearly compared. It is interesting to note here that the SBUV TCO
non-linear trend initially increases from 1998, and then peaks in around 2011, before decreasing.

Frith et al. (2014) found similar behaviour when applying linear trend fits to SBUV TCO, fixing the
start date in January 2000 and incrementally increasing the end date, i.e. the largest positive trend was
found for the period 2000–2011 and thereafter trends decreased. Their analysis ended in 2013, but the
non-linear trend from our DLM analysis, here, shows identical behaviour, and which has continued
decreasing until 2016 and suggests that TCO ozone has now returned to 1998 levels despite an initial

upward trend. Qualitatively similar behaviour is seen in the Merged-SWOOSH/GOZCARDS StCO,
though less pronounced because of its larger overall downward behaviour (see below, section 4.3),
which lends supporting, independent, evidence that such a turnover in ozone trends might be real.
The StCO from Merged-SWOOSH/GOZCARDS continued to decrease after 1998 and, while this
decline stalled in the late 2000s, since 2012 it has continued to decrease. The overall result is that

StCO is on average lower today than in 1998, by ~1.5 DU.

The different stratospheric regimes that contribute to the StCO behaviour can be see in Figs. 3b–d,
where we show, upper, middle (10–32 hPa), and lower stratospheric ozone timeseries from Merged-
SWOOSH/GOZCARDS. A recovery is clear in the upper stratosphere in Fig. 3b, increasing by a
mean of ~1 DU, and trends have been relatively flat since 1998 in the middle stratosphere (Fig. 3c),

with a mean decrease of ~ 0.5 DU. However, the result from Merged-SWOOSH/GOZCARDS in the
lower stratosphere (Fig. 3d) indicates not only that ozone there has declined by ~2 DU since 1998,
and has been the main contributor to the StCO decrease, but that the lower stratospheric ozone has
seen a *continuous and uninterrupted decrease*.

### 4.3 Tropospheric ozone contribution to TCO

The stratosphere accounts for the majority (~90%) of TCO, so intuitively attribution to TCO changes
would be expected to come primarily from this region. However, the results in Fig. 2 and 3 suggest a
discrepancy between StCO and TCO. Despite this, there is no serious conflict between the different
changes indicated by global StCO and TCO distributions (Fig. 2) and trends (Fig. 3a), when the
remaining 10% of the TCO, i.e. tropospheric ozone, is considered, as we show in the following.

First, it is important to establish confidence in the SBUV TCO observations. These have been
very stable since 1998 when comparing SBUV TCO overpass data to the independent ground-based
Arosa TCO observations (Fig. A8). This, therefore, provides confidence in the result that there is
little net change in TCO since 1998. Additionally, Chehade et al. (2014) reported that other TCO
composites agree very well with the SBUV TCO and there is little difference between the various

TCO composites when performing trend analysis.




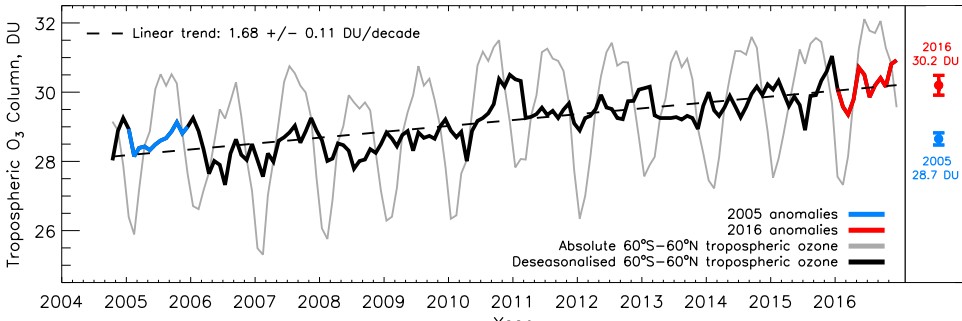

Figure 4: 'Global' 60°S–60°N total tropospheric column ozone between 2004 and 2016. OMI/MLS integrated ozone (grey line) and deseasonalised timeseries (black). The 2005 and 2016 periods are plotted in blue and red, respectively and the mean and two standard errors on the mean for these two years are plotted on the right, with the mean value added alongside. The mean linear trend estimate (dashed line) and the one-standard deviation uncertainty are also provided.

In a second step, we consider global tropospheric ozone changes. In Fig. 4, we present recent estimates from the Aura satellite Ozone Monitoring Instrument and Microwave Limb Sounder (OMI/MLS) instruments of global (60°S–60°N) tropospheric column ozone (TrCO) from 2004 to 2016 (grey), along with deaseasonalised anomalies (solid black); the deseasonalised years 2005 and 2016 are in-

dicated in blue and red – the means (right) indicate a significant increase in ozone. A linear fit to the deseasonalised timeseries indicates an increase in tropospheric ozone of 1.68 DU per decade; if this has held true for the entire 19 year period (1998–2016) it implies a mean increase of ~3 DU, which would more than account for the difference between the StCO and TCO peaks (~1.6 DU) in the upper right panel of Fig. 2.

Supporting evidence for tropospheric ozone increases comes from work reconstructing stratospheric ozone changes in a chemistry climate model (CCM). Shepherd et al. (2014) indicates that tropospheric ozone in the northern (35°–55°N) and southern mid-latitudes (35°–55°S) may have increased by ~1 DU (1998–2011), while equatorial (25°S–25°N) may have increased by ~1.5 DU respectively. While we consider a longer period, this qualitatively agrees with the latitude-resolved

distributions in Fig. 2, which shows that, except for a couple of southern mid-latitudes (30°–40°S and 50°–60°S) and the most northerly band (50°–60°N), all TCO posteriors indicate smaller decreases, or larger increases, compared to the Merged-SWOOSH/GOZCARDS StCO changes.

Returning to the OMI/MLS tropospheric ozone, by separating out TrCO by latitude, and looking at the mean 2005–2015 change shows significant increases in mean TrCO levels at all latitudes, except

a non-significant increase at 50–60S (see Fig A13). The latitudinal structure and magnitude of the TrCO changes, with peaks at ~30° in both hemispheres and clear minima at high latitudes and near



the equator, bears resemblance to the piecewise linear post-1998 TCO trends in Fig. 9 of Chehade et al. (2014) and Fig. 10 of Frith et al. (2014), though there are detailed differences, and so we note that the difference between StCO and TCO in Fig. 2 does not follow this pattern so coherently, which

may be a result of considering a shorter time period for TrCO, and deserves further consideration. OMI/MLS results are not independent from Merged-SWOOSH/GOZCARDS as Aura/MLS forms a part of this composite post-2005. Nevertheless, OMI/MLS is independent from SBUV TCO; the OMI TCO component of the product had a drift of less than 1% per decade with respect to SBUV TCO (McPeters et al., 2015). Regarding OMI, McPeters et al. (2015) stated that the OMI TCO data,

which forms part of the TrCO product when the StCO from Aura/MLS is included, are stable enough for ozone trend studies, that OMI has proven to be one of the most stable instruments flown, and they concluded that OMI provides some of the highest quality ozone data from trend analysis avaliable. Ziemke and Cooper (2017) found no statistically significant drift with respect to various independent measures, or between MLS StCO and OMI StCO residuals, but did detect a small drift of +0.5 DU

per decade in OMI/MLS TrCO caused by an error in the OMI total ozone - this was rectified for the version we consider here.

A deeper investigation is needed to understand difference in the contributions of TrCO and StCO to TCO, especially considering uncertainties carefully, but this is beyond the scope of this work. We note that studies using various data sources show less significant regional increases (and some

decreases) with global estimates ranging from 0.2 to 0.7% per year (~0.6–2 DU per decade) (Cooper et al., 2014; Ebojie et al., 2016; Heue et al., 2016), though these estimates considered different time periods; this suggests a large range of uncertainty, but even the lower end of the estimated increases in TrCO are in line with the missing part of the TCO change, after considering StCO, that we estimate here. Tropospheric ozone is not the main focus of the study here, but the evidence

presented overall suggests the missing component in the declining StCO distributions and trends, with respect to constant TCO, is indeed from increasing tropospheric ozone.

### 4.4   Comparison of stratospheric spatial and partial column ozone trends with models

The observational results for the lower, and whole, stratosphere presented thus far have not been previously reported. However, it is not clear that this represents a departure from our understanding

of stratospheric trends as presented in modelling studies. We present the percentage ozone change from two state-of-the-art chemistry climate models (CCMs) in Fig. 5: (a) the NCAR Community Earth System Model (CESM) Whole Atmosphere Community Climate Model-4 (WACCM; Marsh et al. (2013)); and (b) the SOlar Climate Ozone Links (SOCOL; Stenke et al., 2013) model. Both simulations were performed with the Chemistry Climate Model Initiative phase 1 (CCMI-1) bound-

ary conditions in specified dynamics (SD) mode (see Morgenstern et al. (2017) for information on CCMI and boundary conditions used in models). SD uses reanalysis products to constrain model dynamics towards observations so as to best represent the dynamics of the atmosphere, while leaving





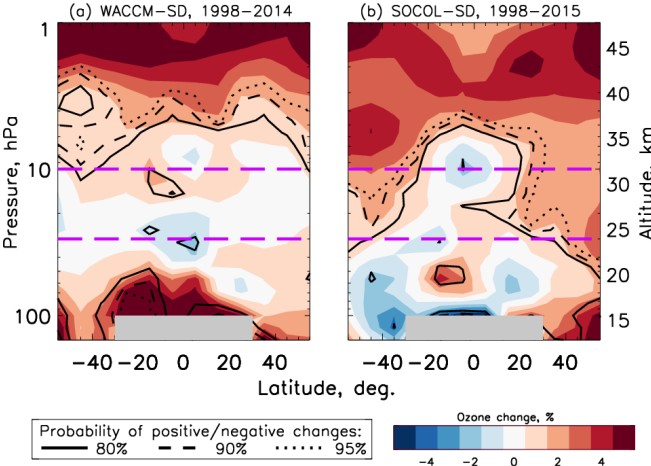

Figure 5: As for Fig. 1, but for (a) WACCM-SD and (b) SOCOL-SD.

chemistry to respond freely to these changes. Such an approach has proven highly accurate at repro-
ducing ozone variability on monthly to decadal timescales in the equatorial upper stratosphere (Ball
et al., 2016). WACCM-SD uses version 1 of the Modern-Era Retrospective analysis for Research and
Applications (MERRA-1; Rienecker et al. (2011) reanalysis[6], while SOCOL-SD uses ERA-Interim
(Dee et al., 2011). Thus, the two models are both independent in terms of how they are constructed,
and the source of nudging fields used, but have similar boundary conditions as prescribed by CCMI-
1.

In Fig. 5 both models display broadly similar behaviour in the upper stratosphere above 10 hPa,
roughly in line with the observations (Fig. 1). Spatially, in the middle stratosphere there are dif-
ferences in sign, but generally significance is low: WACCM-SD displays broadly positive changes
except in the tropics at 10 and 30 hPa, while SOCOL-SD displays a negative spot centred in the
tropics at 10 hPa, while mid-latitudes are often positive and significant. In the lower stratosphere,
SOCOL-SD displays negative trends in the Southern hemisphere lower stratosphere, but positive
in the Northern, while WACCM-SD is generally positive everywhere, and significant at the lowest
altitudes, except at 30–40 hPa in the tropics where a negative tendency is seen. In both SOCOL-SD
and WACCM-SD, trends in the lower stratosphere are generally not significant, and do not display
the clear and significant decreases found in the observations. Posterior distributions similar to those
of Fig. 2 are presented for SOCOL-SD and WACCM-SD in Figs. A9 and A10, respectively. The

---

[6]Use of MERRA-2 reanalysis (Gelaro et al., 2017) makes little difference, except in the upper stratosphere after 2004,
where positive trends are larger when using MERRA-2 (see Fig. A12). The WACCM-SD run with MERRA-2 uses CESM
1.2.2 at 1.9×2.5 horizontal resolution and 88 vertical layers up to 140 km, using prescribed aerosols from the RCP 8.5
scenario.



displayed behaviour is similar to that described here spatially for the models in Fig. 5, and no significant decreases are found (two SOCOL-SD latitude bands display negative changes in the lower stratosphere with ~75% probability: 30–40S and 10–20N). It is worth noting that in both cases the integrated, global trends in the StCO and upper stratosphere are all positive with probabilities of

an increase exceeding 95%, and positive in the lower stratosphere, with 69 and 85% probability of an increase in SOCOL-SD and WACCM-SD, respectively. The non-linear DLM trends (Fig. 3) of WACCM-SD (blue) and SOCOL-SD (purple) emphasize the clearly differing behaviour to the observations, especially in the lower stratosphere (the deaseasonalised and regression model time-series are omitted from Fig. 3 for clarity, but provided in Fig. A11). It is worth mentioning that the

behaviour of TCO from the models was similar to SBUV TCO (Fig. 3a) until around 2012, after which modelled ozone continued to increase while observations show a gradual decline until 2016 (see discussion in section 4.2).

It is notable that also the suite of CCMVal-2 models, the predecessor of CCMI-1, show little significant behaviour in the lower stratosphere, with a tendency for positive changes. As shown in

Fig. 2-10 of the WMO (2014) report the ozone trends in the CCMVal-2 multi-model mean from 2000 to 2013 are positive everywhere except a region bounded by 80–20 hPa and 30S–30N, although only above ~10 hPa are trends significant. Furthermore, there is an ozone increase in the mid-latitude lower stratosphere, albeit non-significant, indicated by the CCMVal-2 models that is not seen in the observations, suggesting that models may not be simulating that region correctly. Extending to 2016

with two independent nudged models, as shown here, does not change this result, which differs from the (i) significant decreases in ozone found in the lower stratosphere, and (ii) the stalled recovery seen in SBUV TCO while models project continued increases.

Chemistry climate models (CCMs) represent our integrated understanding of processes that govern ozone variability and trends, and include chemistry, transport and feedbacks on radiation. Over-

all, they capture the historical behaviour in the stratosphere well (e.g. total column ozone trends driven by EESC changes). However, when it comes to the UTLS region it is not yet clear if models do so well. For example, Figs. 7.27 and 7.28 of the CCMVal-2 report SPARC/WMO (2010) indicate a better model performance with respect to UTLS ozone in summer, when transport effects are weaker and chemistry more important. However, there is a large difference compared to observations

and a wide spread among the models during winter/spring. Transport is affected by many factors, e.g. model vertical/horizontal resolution and gravity wave parameterizations, and trends in atmospheric circulation are also hard to measure and, therefore, to assess the models with. Whether the difference between the models and observations is a result of model design, incorrect boundary conditions (e.g. aerosol contributions from anthropogenic (Yu et al., 2017) or volcanic (Bandoro et al.,

2017) sources may be underestimated), or missing chemistry remains an open question (see below for further discussion).



## 5 Conclusions

In summary, we have presented evidence of highly significant changes in stratospheric ozone between 1998 and 2016. The main findings are that:

(i) the MP is further confirmed to be successfully reducing the impact of hODSs as indicated by the highly probable recovery seen in most upper stratospheric (1–10 hPa / 32–48 km) regions in all composites;

(ii) lower stratospheric ozone (147/100–32 hPa / 13/17–24 km) has continued to decrease since 1998 at all latitudes between 50°S and 50°N;

(iii) there are indications that the total, global (60°S–60°N) stratospheric ozone may have continued to decrease;

(iv) indications of no decrease, or perhaps an increase, in TCO is likely a result of increasing tropospheric ozone, together with the slowed rate of decrease in stratospheric ozone following the MP.

(v) state-of-the-art models, nudged to have historical atmospheric dynamics as realistic as possible do not reproduce the observed decreases in lower stratospheric ozone, which may suggest deficiencies in some aspect of the modelling.

The cause for the continuing decline in lower stratospheric ozone is not fully understood and determining the exact cause is beyond the scope of this study, but there are several possible explana-460 tions. CCM simulations indicate that tropical stratospheric ozone is expected to decrease following increased upwelling in the tropics (<30°) linked to an acceleration of the BDC from greenhouse gas (GHG) induced climate change, which has a larger influence on ozone trends than hODSs in this region (Randel and Wu, 2007; Oman et al., 2010; WMO, 2014); this may account for some of the tropical lower stratosphere ozone decrease, but clear evidence for this in observations remains weak 465 (WMO, 2014). Some modelling and studies also indicate that a rise in the tropopause (Santer et al., 2003), due to the warming troposphere, could lead to a localised ozone decrease (Steinbrecht et al., 1998), though it is not clear of how TCO is affected on large scales, Plummer et al. (2010); Dietmüller et al. (2014); since the troposphere is continuing to warm, the tropopause may continue rising and have an affect on stratospheric ozone. We also pose the hypothesis that an acceleration of the 470 lower stratospheric BDC shallow branch in response to climate change (Randel and Wu, 2007; Oman et al., 2010) may more rapidly transport ozone poor air to the mid-latitudes from the tropical lower stratosphere, where dynamical changes dominate over photochemical ozone production processes (Johnston, 1975; Perliski et al., 1989). While, these possibilities are dynamically-driven responses to climate change, a chemically-driven alternative has also been suggested. Observations indicate an 475 increase in very short lived substances (VSLSs) containing chlorine and bromine species from both



anthropogenic and natural sources (Hossaini et al., 2015). Modelling studies imply that VSLSs preferentially destroy ozone in the lower stratosphere, particularly at mid- and high-latitudes (Hossaini et al., 2015, 2017). It is thought that these species may delay the restoration of the ozone layer to pre-1960s levels, but information is available for only a small number of VSLSs and knowledge of
the reaction rate kinetics to determine their impacts is currently not adequate.

While the reason for the lower stratospheric ozone decline is not yet determined, the signal is clear and the likely consequences significant. The MP is working, but a reduction in harmful UV radiation reaching the surface to pre-1980's levels depends on a restoration of the TCO (WMO, 2014); the lower stratospheric ozone decline appears to be inhibiting this, and models as yet do not reproduce
these downward trends with significance. Increased transport of ozone into the troposphere from the stratosphere is expected if global surface temperatures continue to increase, and may impact air quality (Hegglin and Shepherd, 2009; Neu et al., 2014); current trends suggest that ozone available for such exchange is decreasing. Additionally, ozone in the lower stratosphere is an important factor in radiative forcing (RF) of the climate (Randel and Thompson, 2011), and so far has offset some of
the RF increase from rising GHGs; a reduction in lower stratospheric ozone may lead to reduced RF and further offsetting. Finally, the restoration of the ozone layer is essential to reducing the harmful effects of solar UV radiation on surface life, including humans (Slaper et al., 1996). It is imperative that we determine the cause of the decline in lower stratospheric ozone identified here, both in order to predict future changes, and to determine if it is possible to prevent further decreases.

*Acknowledgements.* GOZCARDS ozone data can be accessed at https://gozcards.jpl.nasa.gov/, SWOOSH at http://www.esrl.noaa.gov/csd/groups/csd8/swoosh/, SBUV-MOD and SBUV TCO at http://acd-ext.gsfc.nasa. gov/Data_services/merged/, and SBUV-MER at ftp://ftp.cpc.ncep.noaa.gov/SBUV_CDR/. W.T.B. and E.V.R. were funded by the SNSF project 163206 (SIMA). We thank the SPARC LOTUS working group as a forum for discussion and data exchange. Work at the Jet Propulsion Laboratory was performed under contract with
the National Aeronautics and Space Administration. GOZCARDS ozone data contributions from Ryan Fuller (at JPL) are gratefully acknowledged. We are grateful to Daniel Marsh and Doug Kinnison for providing ozone data from WACCM CESM in specified dynamics mode.





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





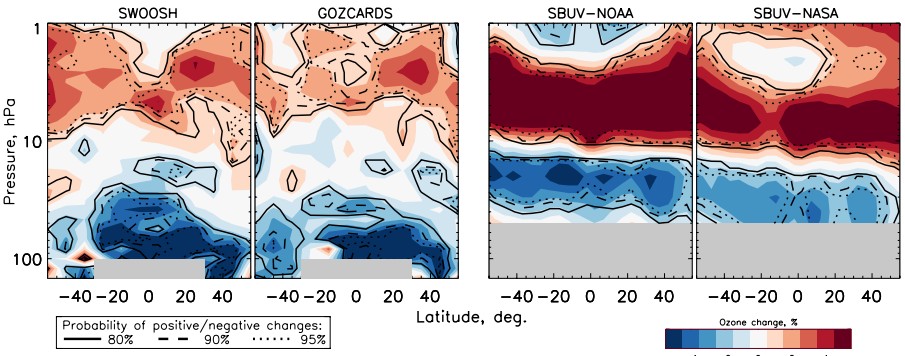

Figure A1: 1998-2016 ozone change. As for Fig. 1; from left to right, SWOOSH, GOZCARDS, SBUV-NOAA, and SBUV-NASA composites.

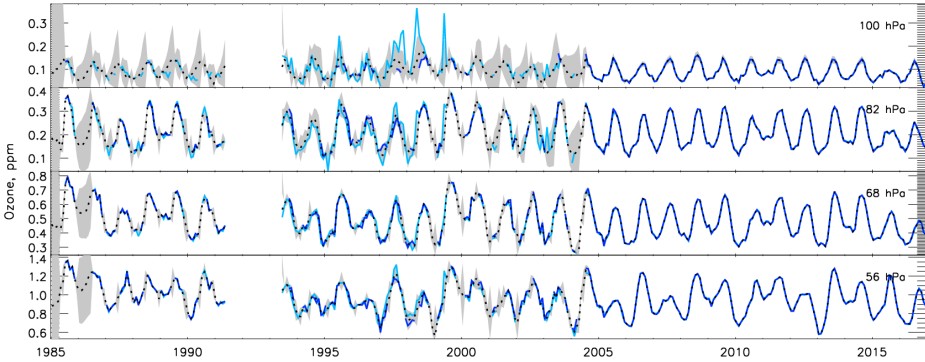

Figure A2: Example of merging result for SWOOSH and GOZCARDS. The result of merging procedure detailed in Ball et al. (2017) applied to GOZCARDS (dark blue), SWOOSH (light-blue) ozone composites in the 0-10°N band at four pressure levels indicated in the top right of each panel. The resulting Merged-SWOOSH/GOZCARDS timeseries is shown as a dashed-black line with two standard deviation uncertainty in grey shading.

Global Modeling Initiative's Chemical Transport Model, Journal of Geophysical Research (Atmospheres), 111, D19303, doi:10.1029/2006JD007089, 2006.



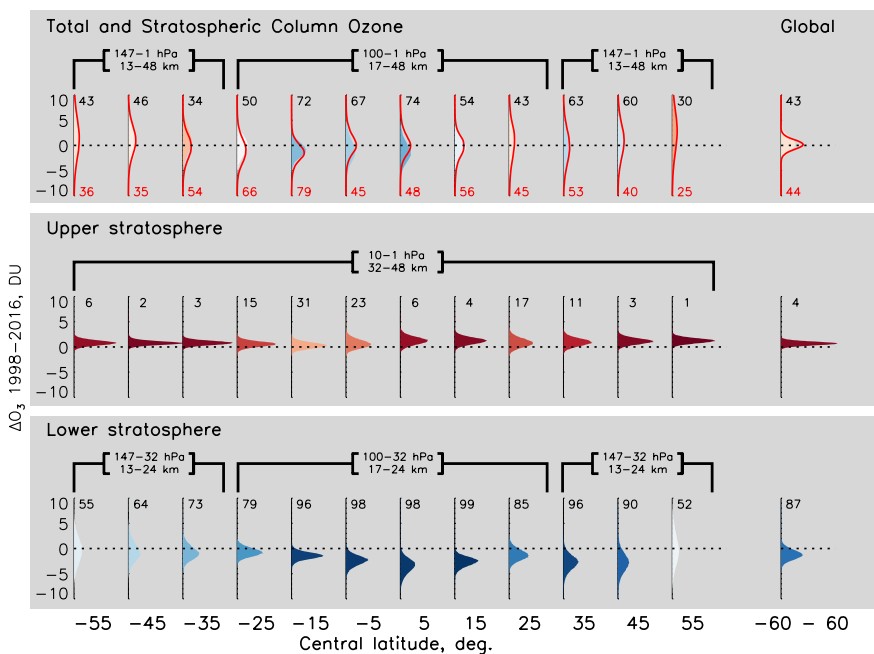

Figure A3: SAGE-II/OSIRIS/OMPS posterior distributions for the 1998-2016 ozone changes. See caption of Fig. 2 for details.





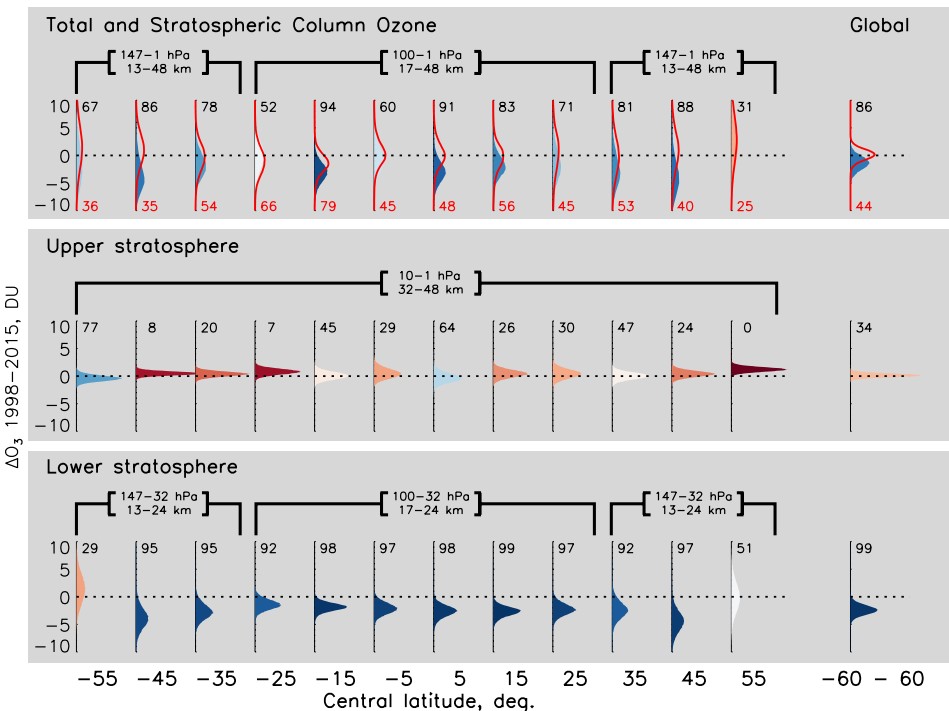

Figure A4: SAGE-II/CCI/OMPS posterior distributions for the 1998-2015 ozone changes. See caption of Fig. 2 for details.





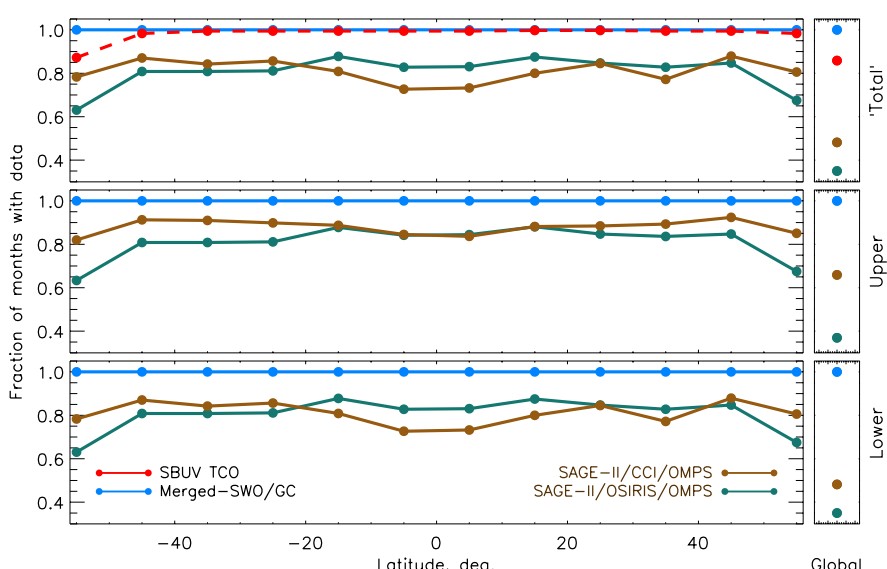

Figure A5: The fraction of data available for the (top) StCO/TCO, (middle) upper stratosphere, and (bottom) lower stratosphere posterior estimates in Figs. 2, A3, and A4. Global (right) are much lower in SAGE-II/OSIRIS/OMPS and SAGE-II/CCI/OMPS because if data are missing in any latitude in a particular month, the global PCO or StCO is assigned no data.



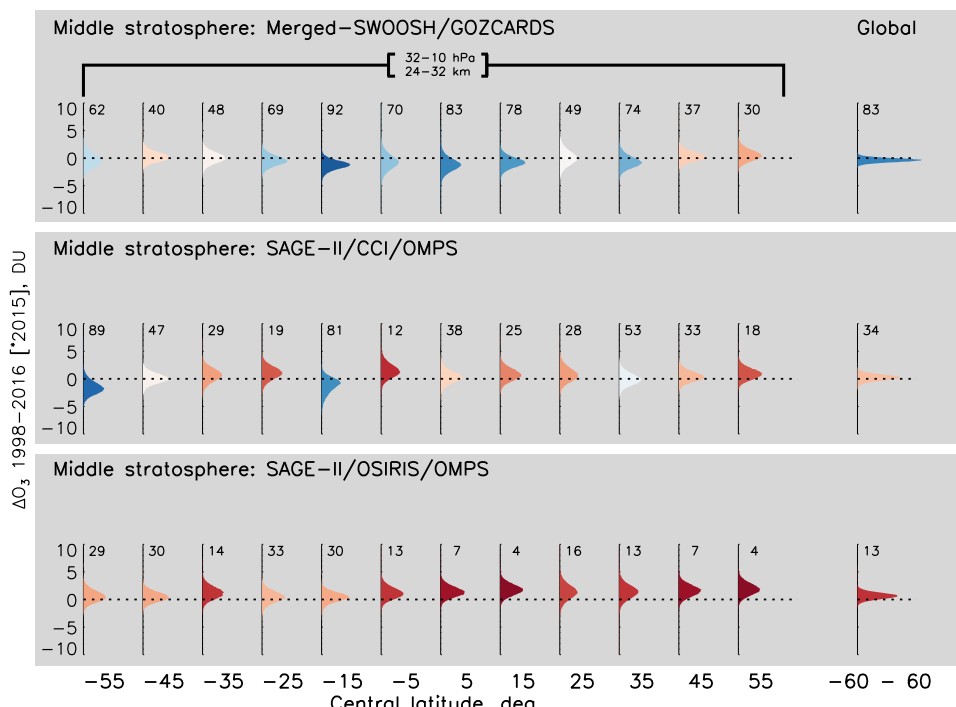

Figure A6: Posterior distributions in the middle stratosphere for the 1998-2015/2016 ozone changes. Similar to panels in Fig. 2, but for the middle stratosphere (32-10 hPa, ~24-32 km) for (top) Merged-SWOOSH/GOZCARDS, (middle) SAGE-II/CCI/OMPS, and (bottom) SAGE-II/OSIRIS/OMPS.





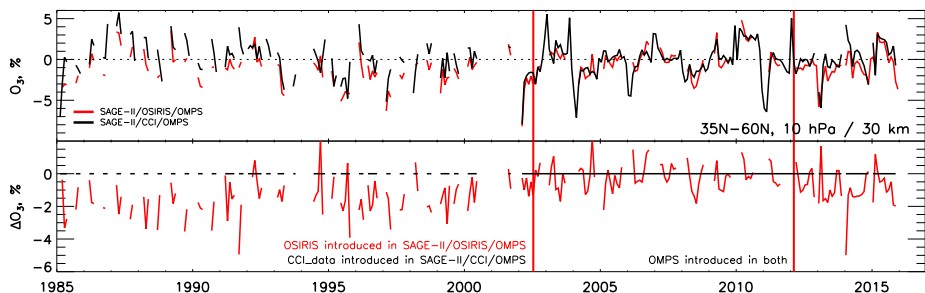

Figure A7: SAGE-II/OSIRIS/OMPS (red) and SAGE-II/CCI/OMPS (black) at northern mid-latitudes at 30 km. (Upper) The deseasonalised changes relative to 2005–2013, and (lower) SAGE-II/OSIRIS/OMPS minus SAGE-II/CCI/OMPS. Approximate dates when OSIRIS and OMPS were introduced into the composites are shown with vertical red lines, before/after which a shift in the mean appears.

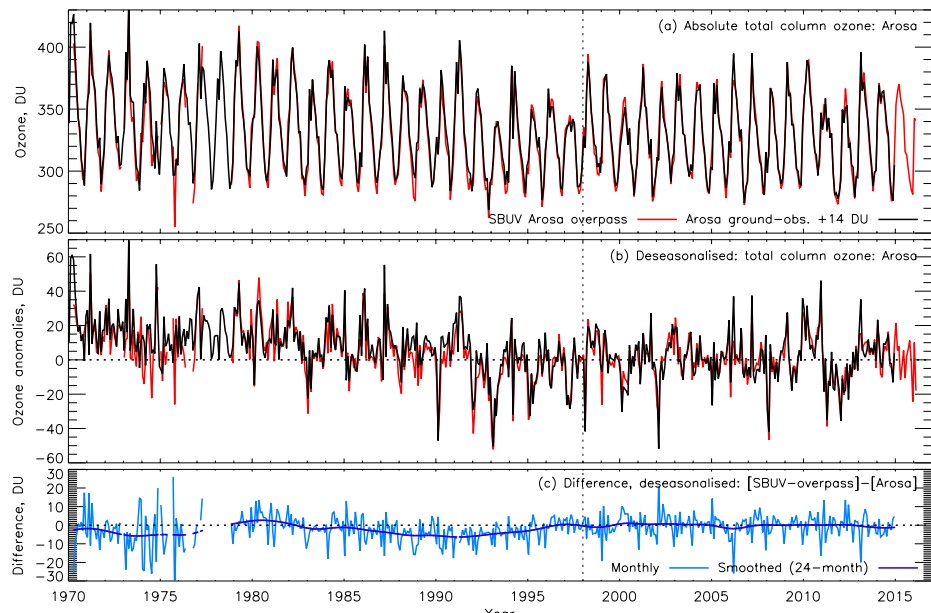

Figure A8: Ozone over Arosa, Switzerland, 1970-2016. (a) Absolute ozone from SBUV Total Column Ozone (TCO) overpass observations (red) compared to the ground-based Arosa TCO observations (black), with Arosa shifted to the SBUV mean for 1998-2013; (b) as for (a) with the seasonal cycle removed; (c) the monthly difference between timeseries in (b) and a 24-month Gaussian smoothing (thick line).





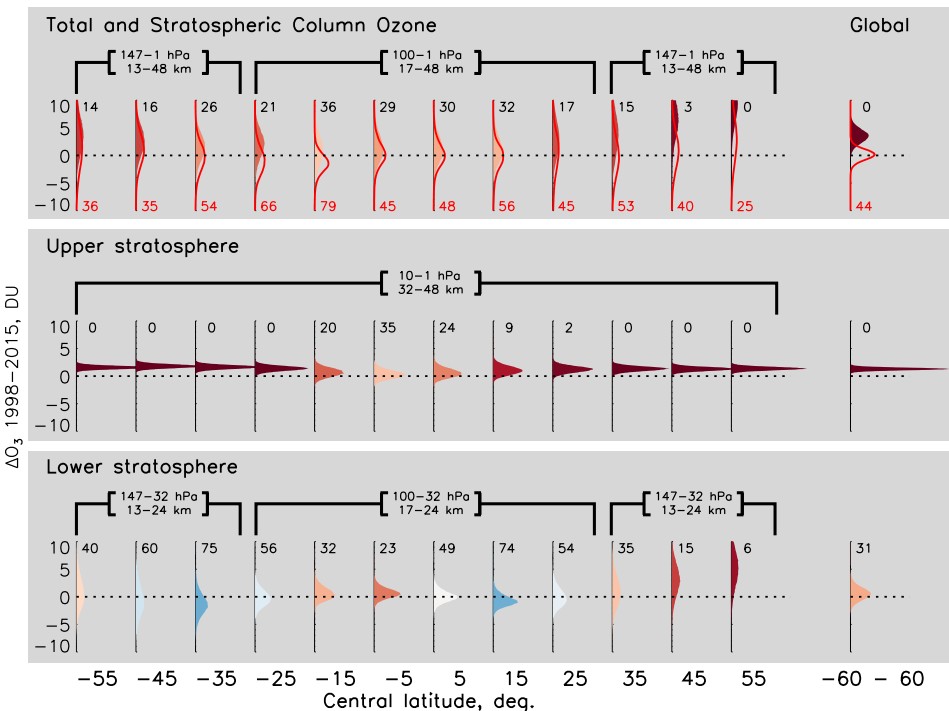

Figure A9: SOCOL-SD posterior distributions for the 1998-2015 ozone changes. See caption of Fig. 2 for details.





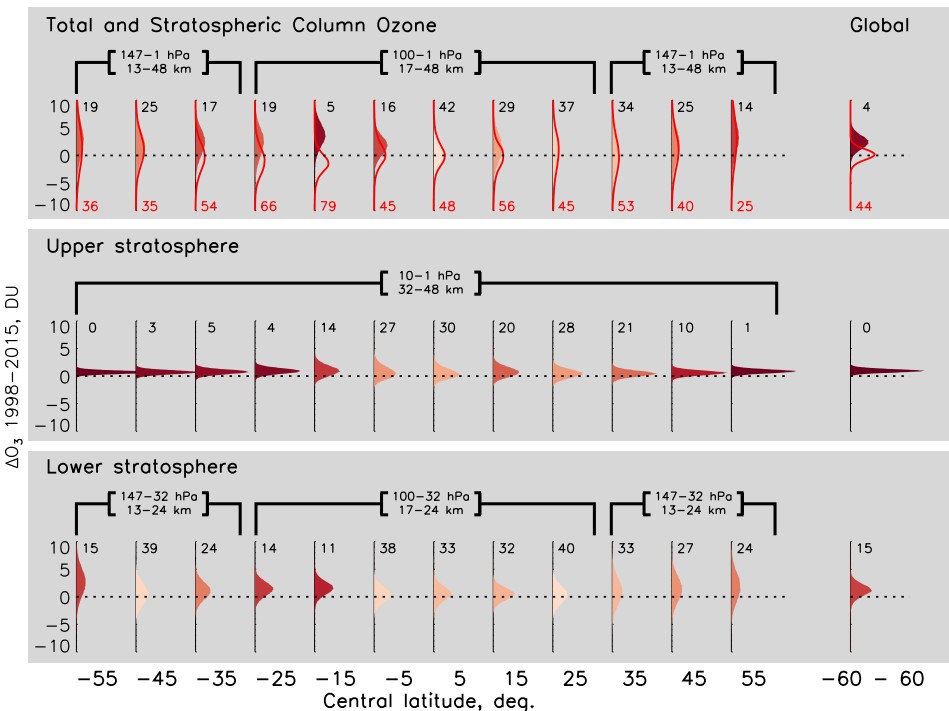

Figure A10: WACCM-SD posterior distributions for the 1998-2014 ozone changes. See caption of Fig. 2 for details.





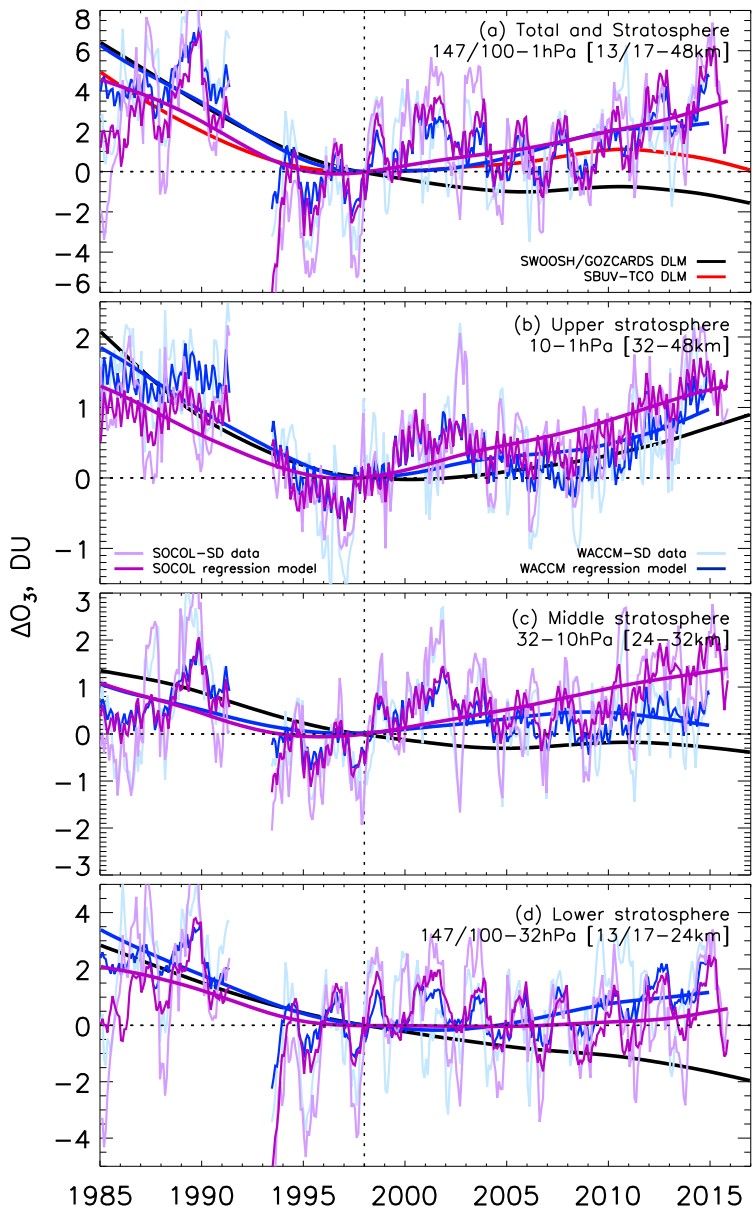

Figure A11: As for Fig. 3, but with deseasonalised and regression model timeseries from SOCOL-SD (purple) and WACCM-SD (blue). DLM results for SBUV-TCO and Merged-SWOOSH/GOZCARDS are retained in this plot from Fig 3; see Fig 3 for more details.





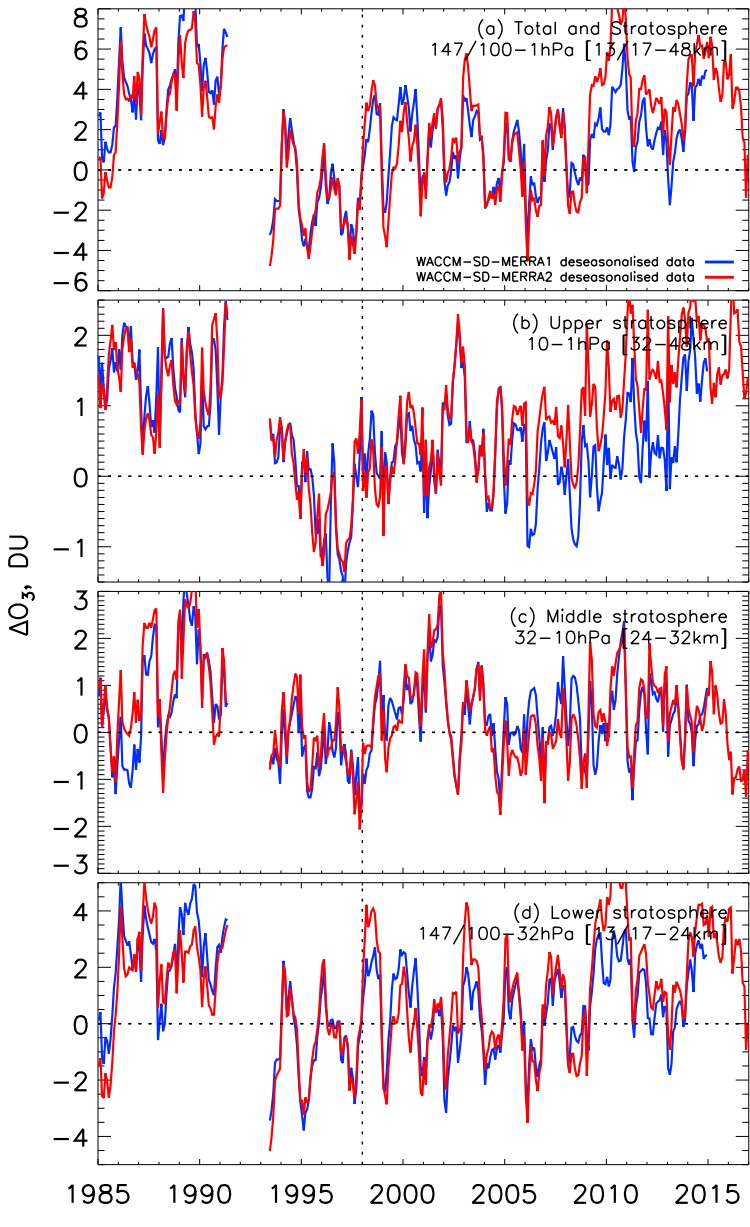

Figure A12: As for Fig. 3 and A11, but with deseasonalised model timeseries only from WACCM-SD using MERRA-1 (blue) and MERRA-2 (red).




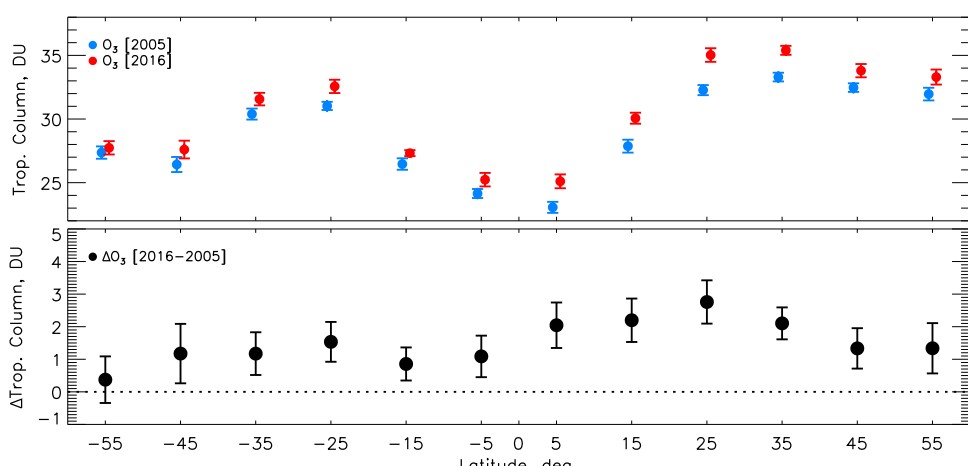

Figure A13: Mean and two standard errors of tropical column ozone (TrCO) change between 2005 and 2016 from OMI/MLS. The upper panel shows the absolute levels in 10° latitude bins in 2005 (blue) and 2016 (red), while the lower panel gives the difference between 2005 and 2016 with combined errors, similar to the right panel of Fig. 4.