# Peer review of "Evidence for a continuous decline in lower stratospheric ozone offsetting ozone layer recovery"

_Atmospheric Chemistry and Physics, 2017_

## Referee Comment (RC1) · Anonymous Referee #2 · 23 Oct 2017

The authors use a new dynamical linear modelling method to identify slowly varying trends in the global ozone profile. They find increasing ozone in the upper stratosphere since the late 1990s, little change in mid-stratospheric ozone since the 1990s, but significantly declining lower stratospheric ozone over the entire 1985 to 2016 period. These results are generally consistent with a number of recent ozone trend studies. However, this study is the first to focus on the lower stratospheric decline, whereas many other studies do not show significant decline in the lower stratosphere, and/or do not focus on this region. The decline in lower stratospheric ozone would explain why, so far, no significant increases in total column ozone have been observed, despite the decline of ozone depleting substances since the late 1990s, and despite expectations from model simulations. Model simulations, in fact, indicate that lower stratospheric

ozone should be increasing. As pointed out by the authors, a decline in lower stratospheric ozone, as reported here, seriously questions our understanding of global ozone trends, and our ability to model them.

**1 General Comments**

Overall, this is a good paper, well suited for ACP, and deserving publication. There are aspects, however, where I am not certain, and where I feel a bit more scepticism would be appropriate.

1. Originators of the merged SBUV (Frith et al., 2017, https://doi.org/10.5194/acp-2017-412), CCI (Sofieva et al. 2017, https://doi.org/10.5194/acp-2017-598) and SOO (Bourassa et al. 2017, https://doi.org/10.5194/amt-2017-229) data sets do not have the same confidence as the authors into the stability and reliability of their ozone records in the lowermost stratosphere. Frith et al., (2017) do not report trends below 30 hPa (25 km), Sofieva et al. (2017) do not report trends below 20 km (50 to 70 hPa). Bourassa et al. (2017) do not report trends below 18 km (70 hPa). Given this, more caution on the reliability of lower stratospheric ozone (147/100 to 32 hPa; 13/16 to 27 km) in this study would be appropriate. Uncertainties in this region are large, and easily exceed 5% (see e.g. Fig. 9 of Sofieva et al.). With such low accuracy, small trends like the one reported here (-2 DU / decade, for a value of maybe 100 DU), of the order of a few percent per decade, are always questionable, and have to be put into perspective.

2. Figure 3 demonstrates steps in the lower stratospheric ozone time series. These might be due to instrumental changes relevant for all merged data sets. The large downward step by about 2 DU in 2004/2005 occurs at exactly the time when the SWOOSH and GOZCARDS merged ozone records switch from the very sparsely

sampling solar occultation SAGE II instrument (operating until 2005, with even more reduced sampling since 2001) to the very densely sampling Microwave Limb Sounder (since 2004), an instrument with characteristics very different from SAGE II. Similar things apply for the CCI data set at the switch-over from SAGE II to ENVISAT instruments around 2002/2003, where, e.g., Figs. 8 and 9 from Sofieva et al. (2017) demonstrate the large changes in sampling and the large uncertainties in the lowermost stratosphere. These changes could be very important for time series and for trends in the lowermost stratosphere. I think the authors should add some more caution here. How do the curves look for CCI and SOO?

3. The tropospheric OMI/MLS column in Fig. 4 does not provide an independent piece of information. It just shows that the difference between OMI total column ozone, which should be very similar to SBUV total column in the present study (and have no trend since about 2000), and MLS stratospheric column (essentially the same as SWOOSH and GOZCARDS used in the present study) has a positive trend. Since upper stratospheric is increasing, this just means that lower stratospheric ozone from MLS (and GOZCARDS, SWOOSH) must be decreasing. While this confirms the findings of the authors, it still hinges on the same MLS data, and does not provide an independent piece of information. Independent information about tropospheric ozone trends must come from somewhere else. However, recent studies show no trend for zonal mean tropical tropospheric ozone based on GOME/SCIAMACHY/GOME2 data (Leventidou et al., 2017, https://doi.org/10.5194/acp-2017-815), or provide little confidence on our ability to identify large scale tropospheric ozone trends (e.g. Cooper et al. 2014, http://doi.org/10.12952/journal.elementa.000029).

I acknowledge that in parts of the manuscript, the authors are mentioning these open questions. However, I do feel that they should be a more integral part of the manuscript. Therefore, I suggest that the authors reword / change parts of their manuscript, to better

reflect these open questions. Below, I'll indicate in more detail which specific parts I am talking about.

**2 Detailed Comments**

Title: Given all the uncertainties, I would put a question mark behind the title.

Lines 6, 13, 14, ...: I find the abbreviations TCO, StCO, TrCO unnecessary and annoying. Every time I read them, I have to re-think what is meant. I would prefer to have them spelled out, throughout the manuscript: total column ozone, stratospheric column ozone, tropospheric column ozone. Text length would not change.

Lines 14-15: Delete "and harmful to respiratory health". This is irrelevant in the context of the paper. In fact, given the uncertainties mentioned above, I think the entire sentence about tropospheric ozone increase could be omitted, or at least reworded. Certainly, tropospheric ozone changes are not investigated thoroughly in the present paper.

Lines 17 to 20: Not investigated in the paper. The last sentence should be removed.

Line 30: I think a reference is required here.

Lines 32-33: I think we are far from attribution in the IPCC sense. Therefore I would suggest to delete "an attribution", insert "due" before "to decreasing ODS", and replace "possible" by "reported."

Line 34: after "rates" add "and by accelerating ozone transport through the meridional Brewer Dobson Circulation".

Line 36: A reference is needed here.

Lines 37 to 97: This is quite longish and wordy, and seems to have been written in several steps and at different times. I would recommend to shorten and compact this: The paragraph about total ozone (around line 40) should include the newest results from Weber et al. (2017, https://doi.org/10.5194/acp-2017-853). The part about differences between MLR, EESC, PWLT (lines 42 to 60) should be moved to the end (line 98), and should be shortened and combined with the paragraph starting in line 98. The paragraph around line 90 should mention more about the general uncertainties of ozone measurements in the lower-most stratosphere, see also my general remark above. Overall, I think the entire introduction could be shortened by 20 to 30%, because many things are clear to an ACP audience, and are also mentioned again later.

Line 65: Frith et al., 2017, https://doi.org/10.5194/acp-2017-412, should be added here.

Line 77: This could/ should also include relevant references from lines 64, 65.

Line 84: I think this is a key point here: Instrumental uncertainties are 10 to 15%, and the "observed" lower stratospheric ozone decline is only about 2 DU out of maybe 100 DU. Can we believe a 2% effect measured by a system that is only accurate to within 10 or 15% ?

Lines 86-87: There are good reasons, why many of the data providers do not trust derived trends below 18 to 20 km. See e.g. Fig. 9 of Sofieva et al. (2017).

Line 100: The work by Damadeo et al. (2014, https://doi.org/10.5194/acp-14-13455-2014; 2017, https://doi.org/10.5194/acp-2017-575) should be referenced as well.

Line 106, 107: It is no big achievement to not report ozone changes as percentages. Suggest to drop ", i.e. . . . in percentage"

Line 115: The sentence does not make sense. Something is missing here.

Line 121: Replace "trends have been" by "about ozone trends"?

Lines 125 to 140: Reduce duplications with what has already been said in lines 42 to 60.

Line 139: Probably better to say "PWLT" instead of "linear trend".

Line 144: What is the correlation between F30 and F10.7 on the time steps used in the present analysis? What is the correlation of the two proxies with ozone, and are there any significant differences between F30 and F10.7 for this type of ozone trend analysis?

Line 157: Delete "being considered"?

Lines 166 to 170: Since a lot of these data sets have changed recently, e.g. from Tummon et al., 2015 to Steinbrecht et al. 2017, I think it is absolutely necessary to indicate already here which data and versions were in fact used. This may require a small table. Mentioning the SPARC LOTUS initiative, which brought together many of the datasets, would also be a good thing.

Lines 185, 187: I think Frith et al. (2017) needs to be added here.

Lines 191 to 199: It would be better to drop this here, and include the relevant information into the paragraph from lines 166 to 177.

Lines 206 to 213: Given my major comment above, and the general question about relevance / independent information content of the OMI-MLS data set: Maybe drop the entire paragraph? I think a short mention in the description of Fig. 4 would be enough. Only if the authors decide to make a stronger point about tropospheric increases, e.g., by adding an analysis of ozone trends from ozone sounding stations, then a separate sub-section would be appropriate.

Line 239: As mentioned in my major comments, I am still only ≈90% convinced that ozone has declined in the lower stratosphere. Therefore, I suggest to replace "clearly indicate" by "give a strong indication".

Line 250: I think more words of caution about the high variability of ozone in the low-ermost stratosphere, and about the poorer accuracy of the measurements there (compared to the mid- and upper stratosphere) would be required here. See also Fig. 9 of Sofieva et al. 2017.

Around lines 300, 327: Also compare with / better compare to Weber et al. 2017.

Line 310: Again: This fairly small change by 1.5 DU is challenging the limited accuracy of the instruments, which is around 1% or 3 DU for total column ozone, and around 5 to 10% for the lowermost stratosphere (= 2 to 5 DU, assuming 50 DU sit in the lower stratosphere). A large part of the observed 2 DU drop in the lower stratosphere around 2004/5 hinges on poorly sampled data from SAGE II, at the end of its lifetime.

Lines 331 to 366: Given my major comments about the OMI/MLS tropospheric ozone results, and in favor of conciseness of the paper: Would it not be much better to drop much of this discussion, drop Figs. 4 and A13? Instead just mention possible tropospheric ozone increases from OMI/MLS and other, more independent sources of information and put them into perspective. Essentially, this could be done with an expansion of the paragraph in lines 367 to 376. The main messages of the paper would remain. Questionable information would disappear, and conciseness would be improved.

Lines 418 to 427: Is this paragraph necessary? I think it could easily be dropped. The entire section 4.4 is quite long and wordy. I think it could be shortened and made more concise.

Line 452: Here is one place, out of many, where TCO left me very confused. I was thinking of TCO = tropospheric column ozone, and saw little sense in the paragraph. As mentioned, spelling out TCO, StCO, TrCO, ... would help readers like me.

Line 475: If you do the numbers, this is still a very small effect for past total ozone columns, maybe 0.2 DU per decade. I think this should be said here.

Lines 458 to 494: Again, I think this is quite long and wordy, and would benefit from

substantial shortening.

Figure A7: Can you show similar plots for the altitudes where it really matters, e.g. 18 km? And also include SWOOSH / GOZCARDS?

To summarize again: I think this is a good paper. I think it would benefit greatly from addressing my major points raised above. It would also benefit substantially from fleshing out redundancies and shortening the text. When this has been done, I fully recommend publication.

---

## Referee Comment (RC2) · Anonymous Referee #1 · 3 Nov 2017

The manuscript "Continuous decline in lower stratospheric ozone offsets ozone layer recovery" by Ball et al. describes analyses of vertically resolved stratospheric ozone data sets of different origin with regard to detection of ozone recovery. For this, a relatively new method in ozone analyses, Dynamical Linear Modelling, is used. Obtained results indicate an increase in upper stratospheric ozone, especially in the mid-latitudes, and a decrease in lower stratospheric layers, especially in the mid-latitudes and tropics. The stratospheric profiles of the different data sets are then integrated to partial columns to analyze the different trend behavior in more detail. Results are compared to tropospheric ozone time series and results of two chemistry-climate model simulations (calculated in specified dynamics mode) to better understand the lower stratospheric ozone trends in particular.

[Figure]

The structure of the manuscript is clear, and it is very well written. The applied methods are described mostly in sufficient detail to allow the reader to understand what was done. It is also stated with plenty of references from the recent literature where this study compares to previous findings, and where new results are presented. There are a few minor things that I would like the authors to address (mainly clarifications, shortening/expansion of explanations, etc.) before I would recommend the manuscript for publication.

General suggestions/comments:

- There are several acronyms that are specified multiple times throughout the manuscript (StCO, PCO, TrCO...). In most cases this is not necessary, but only slightly annoying for the reader. I would suggest either using the full name throughout the manuscript (if the authors think that the reader might not remember the acronym), or defining them once and using them from thereon.

- In some cases throughout the manuscript the authors could be slightly more specific when describing something, e.g.: page 2, line 34 'Models predict...' -> what kind of models?; page 3, line 37 'Only recently has a TCO recovery been detected during the austral spring...' -> the recovery was detected in Antarctica, which is not necessarily deductible from the description; page 4, line 106 'Our aim here is to quantify the absolute changes in ozone...' -> which ozone is referred to here? Stratospheric or tropospheric, global or specific latitude bands? I would suggest that the authors check the manuscript carefully to make sure all descriptions are detailed enough so that it is clear what is described.

- Some lines in the contour plots (e.g. Figure 1, Figure 5, Figure A1...) are hard to see if the contour colors are very dark. If that is the case, maybe the contours for the probability changes (that are black now) could be white instead? That might help them having better visibility.

Specific comments:

- Page 1, title: I think the title is not precise enough. I would suggest changing 'ozone layer recovery' to 'total column ozone recovery' (or something along those lines). As far as I understood, that was the focus of the study.

- Page 4, line 95: after the parenthesis, 'km' is too much

- Page 6, section 2.3: This section is too brief in my opinion. It is not clear how exactly the DLM works, and how the probabilities are calculated. I don't think the explanations have to go into too much detail, but some more explanations would be great.

- Page 7, line 188: '. . .developed by (Ball et al., 2017), . . .' -> parenthesis are placed wrong

- Page 17, line 418-441: The comparisons between the CCMVal-2 results are too lengthy and in some aspects unnecessary. I think these paragraphs could be shortened quite a bit.

- Page 18, Section 5: The conclusion section starts slightly abrupt in my opinion. It would be good to start with some perspective again: where do the findings fit in the bigger picture? What exactly did the authors want to present? Starting with this, it would be easier for the reader to follow the summary of the results that are given with the Roman numberings.

- Page 18, line 467-468: parenthesis for the references Plummer et al. (2010) and Dietmüller et al. (2014) seem wrong

- Page 19, last paragraph: The list of positive effects of the lower stratospheric ozone decrease (decreasing exchange with troposphere, radiative forcing offset, etc.) comes across a little too strong compared to the reasoning why the decline

could be bad for life on Earth. The authors might want to think about rewording some of it to strengthen the point why the decline in stratospheric ozone might indeed be not so good.

- Page 19, line 485: 'trends' should be 'trend'? After all, it is only the lower stratospheric ozone that shows that decline

---

## Author Comment (AC1) · 11 Dec 2017

**Response to referee comments on "Continuous decline in lower stratospheric ozone offsets ozone layer recovery" by W. T. Ball et al**

**General comments relevant to both referees:**

We thank both reviewers for their useful input that has led to clarifications of issues, particularly related to uncertainties, and to a streamlining and improved manuscript. Please see our comments (blue) below in response to the reviewers (black). Any major changes to the text (see below) have been put in bold font in the updated manuscript.

One point worth mentioning is that the method used to merge and account for artefacts in the composites, i.e. in Merged-SWOOSH/GOZCARDS and Merged-SBUV, was based upon an approach that was detailed in the review stage manuscript of Ball et al., 2017 (ACPD), which is now published in ACP. The final version of that paper changed some details in the merging algorithm, which also improved it. There was little affect on the overall result, and there are no changes in the conclusions, but some of the numbers/confidence levels within this manuscript currently under review have changed slightly (i.e Fig 1, 2 and 3). Most notably, the 92% probability of Merged-SWOOSH/GOZCARDS showing a decline in 'global' stratospheric ozone has increased to 95%.

Following comments from both reviewers regarding the title, it has been changed to: "Evidence for a continuous decline in lower stratospheric ozone offsetting ozone layer recovery".

**Anonymous Referee #1**

The manuscript "Continuous decline in lower stratospheric ozone offsets ozone layer recovery" by Ball et al. describes analyses of vertically resolved stratospheric ozone data sets of different origin with regard to detection of ozone recovery. For this, a relatively new method in ozone analyses, Dynamical Linear Modelling, is used. Obtained results indicate an increase in upper stratospheric ozone, especially in the midlatitudes, and a decrease in lower stratospheric layers, especially in the mid-latitudes and tropics. The stratospheric profiles of the different data sets are then integrated to partial columns to analyze the different trend behavior in more detail. Results are compared to tropospheric ozone time series and results of two chemistry-climate model simulations (calculated in specified dynamics mode) to better understand the lower stratospheric ozone trends in particular. The structure of the manuscript is clear, and it is very well written. The applied methods are described mostly in sufficient detail to allow the reader to understand what was done. It is also stated with plenty of references from the recent literature where this study compares to previous findings, and where new results are presented. There are a few minor things that I would like the authors to address (mainly clarifications, shortening/expansion of explanations, etc.) before I would recommend the manuscript for publication. General suggestions/comments:

- There are several acronyms that are specified multiple times throughout the manuscript (StCO, PCO, TrCO. . .). In most cases this is not necessary, but only slightly annoying for the reader. I would suggest either using the full name throughout the manuscript (if the authors

think that the reader might not remember the acronym), or defining them once and using them from thereon.

All abbreviations of the TCO, PCO, StCO, and TrCO variety have been written out in their full form.

- In some cases throughout the manuscript the authors could be slightly more specific when describing something, e.g.: page 2, line 34 'Models predict. . .' -> what kind of models? - page 3, line 37 'Only recently has a TCO recovery been detected during the austral spring. . .' -> the recovery was detected in Antarctica, which is not necessarily deductible from the description; page 4, line 106 'Our aim here is to quantify the absolute changes in ozone. . .' -> which ozone is referred to here? Stratospheric or tropospheric, global or specific latitude bands? I would suggest that the authors check the manuscript carefully to make sure all descriptions are detailed enough so that it is clear what is described.

We have looked through the manuscript and changed as we saw ambiguous. Specific to the referee's suggestions, we have changed: 'models' to '**Chemistry climate** models **(CCMs)**...'; 'been detected during austral spring' → 'been detected over **Antarctica** during austral spring'; 'Our aim here ... ozone...' → '**Here, we quantify the absolute changes in ozone in different regions of the stratosphere, and troposphere, and their contributions to total column ozone, at different latitudes and globally, since 1998**...'

- Some lines in the contour plots (e.g. Figure 1, Figure 5, Figure A1. . .) are hard to see if the contour colors are very dark. If that is the case, maybe the contours for the probability changes (that are black now) could be white instead? That might help them having better visibility.

Agreed. White also has a similar (but opposite) effect at the interface between positive and negative, so after many tests, we settled on a darkish grey.

Specific comments:

Page 1, title: I think the title is not precise enough. I would suggest changing 'ozone layer recovery' to 'total column ozone recovery' (or something along those lines). As far as I understood, that was the focus of the study.

The referee makes a good point and we have considered this carefully. However, this is a little tricky, since the total column ozone recovery is not just the stratosphere. It appears that a significant portion 'may' be due to tropospheric increases, and then the recovery should not be attributed to the total column since it really refers to stratospheric ozone. Since it appears to be the case (whichever of the datasets analysed) that the lower stratosphere is decreasing in such a way that it compensates the ozone layer recovery and the total column ozone increase, we suggest the title represents the more confident result of the stratosphere itself.

Page 4, line 95: after the parenthesis, 'km' is too much

Done.

Page 6, section 2.3: This section is too brief in my opinion. It is not clear how exactly the DLM works, and how the probabilities are calculated. I don't think the explanations have to go into too much detail, but some more explanations would be great.

The DLM approach is explicitly detailed in Laine et al., 2014, and is too detailed to be expanded upon here. Nevertheless, we have added the following to section 2.1:
"We infer posterior distributions on the non-linear trends by Markov Chain Monte Carlo (MCMC) sampling; **the background trend levels at every month are included as free parameters, with a data-driven prior on the smoothness of the month-to-month trend variability. DLM analyses have more principled uncertainties than MLR since they are based on a more flexible model, and formally integrate over uncertainties in the regression coefficients, (non-stationary) seasonal cycle, autoregressive coefficients and parameters characterizing the degree of non-linearity in the trend.** The time-varying, background changes are inferred, rather than specified by [...]"

Section 2.3 has been restructured, we have added the following text to elaborate on how probabilities were estimated:
**"The posterior distributions that represent the change since January 1998 are formed from the (n=100,000) DLM samples from the MCMC exploration of the model parameters (see section 2.1). Then, probability density functions (PDFs) are estimated as histograms of the sampled DLM changes from 1998. Finally, the probabilities represent the percentage of the posterior samples that are negative; therefore, the posteriors and probabilities presented in all figures represent the full information inferred about the change in ozone since 1998 obtained from the DLM analysis; these are not always normally distributed."**

Page 7, line 188: '. . .developed by (Ball et al., 2017), . . .' -> parenthesis are placed wrong

Fixed.

Page 17, line 418-441: The comparisons between the CCMVal-2 results are too lengthy and in some aspects unnecessary. I think these paragraphs could be shortened quite a bit.

Following this, and the second reviewer's suggestion, these paragraphs have been significantly shortened and merged. It now reads:
**"The CCMVal-2 multi-model-mean 2000-2013 ozone changes in the WMO 2014 ozone assessment (Fig. 2-10) show a positive, but insignificant, change in the lower stratosphere at mid-latitudes, which suggests models may not be simulating that region correctly, consistent with the two models extended to 2016 here. While CCMs capture historical ozone behaviour in the upper stratosphere well, it is less clear in the UTLS region. Figs. 7.27-7.28 of the SPARC (2010) report indicate large differences compared to observations in winter/spring, perhaps related to factors affecting model transport (e.g. resolution, and gravity wave parameterizations). Whether these differences result from model design, incorrect boundary conditions (e.g. underestimated anthropogenic (Yu et al., 2017) or volcanic (Bandoro et al., 2017) aerosol contributions), or missing chemistry remains an open question."**

Page 18, Section 5: The conclusion section starts slightly abrupt in my opinion. It would be good to start with some perspective again: where do the findings fit in the bigger picture? What exactly did the authors want to present? Starting with this, it would be easier for the reader to follow the summary of the results that are given with the Roman numberings.

We have added the following at the beginning of the conclusions section: "**Following the successful implementation of the Montreal Protocol (MP), total column ozone stabilised at the end of the 1990s, and searches for the first signs of recovery in total column ozone have been underway since then (Weber et al., 2017; Chipperfield et al, 2017). We find that counteracting trends within different atmospheric layers are the reason a significant detection has remained elusive.** In summary..."

Page 18, line 467-468: parenthesis for the references Plummer et al. (2010) and Dietmüller et al. (2014) seem wrong

Fixed.

Page 19, last paragraph: The list of positive effects of the lower stratospheric ozone decrease (decreasing exchange with troposphere, radiative forcing offset, etc.) comes across a little too strong compared to the reasoning why the decline could be bad for life on Earth. The authors might want to think about rewording some of it to strengthen the point why the decline in stratospheric ozone might indeed be not so good.

We have reduced the strength of the positive benefits and shortened the paragraph overall, which heightens the more negative consequences of a decreasing ozone layer.

Page 19, line 485: 'trends' should be 'trend'? After all, it is only the lower stratospheric ozone that shows that decline

Agreed, and updated.

References:

Chehade, W., Weber, M., and Burrows, J. P.: Total ozone trends and variability during 1979-2012 from merged data sets of various satellites, Atmospheric Chemistry & Physics, 14, 7059–7074, doi:10.5194/acp-14-7059-2014, 2014.
Dudok de Wit et al., J. Space Weather Space Clim 4 (2014) A06, 10.1051/swsc/2014003
McPeters, R. D., Frith, S., and Labow, G. J.: OMI total column ozone: extending the long-term data record, Atmospheric Measurement Techniques, 8, 4845–4850, doi:10.5194/amt-8-4845-2015, 2015.
Weber, M., Coldewey-Egbers, M., Fioletov, V. E., Frith, S. M., Wild, J. D., Burrows, J. P., Long, C. S., and Loyola, D.: Total ozone trends from 1979 to 2016 derived from five merged observational datasets – the emergence into ozone recovery, Atmos. Chem. Phys. Discuss., 2017.

---

## Author Comment (AC2) · 11 Dec 2017

**Response to referee comments on "Continuous decline in lower stratospheric ozone offsets ozone layer recovery" by W. T. Ball et al**

**General comments relevant to both referees:**

We thank both reviewers for their useful input that has led to clarifications of issues, particularly related to uncertainties, and to a streamlining and improved manuscript. Please see our comments (blue) below in response to the reviewers (black). Any major changes to the text (see below) have been put in bold font in the updated manuscript.

One point worth mentioning is that the method used to merge and account for artefacts in the composites, i.e. in Merged-SWOOSH/GOZCARDS and Merged-SBUV, was based upon an approach that was detailed in the review stage manuscript of Ball et al., 2017 (ACPD), which is now published in ACP. The final version of that paper changed some details in the merging algorithm, which also improved it. There was little affect on the overall result, and there are no changes in the conclusions, but some of the numbers/confidence levels within this manuscript currently under review have changed slightly (i.e Fig 1, 2 and 3). Most notably, the 92% probability of Merged-SWOOSH/GOZCARDS showing a decline in 'global' stratospheric ozone has increased to 95%.

Following comments from both reviewers regarding the title, it has been changed to: "Evidence for a continuous decline in lower stratospheric ozone offsetting ozone layer recovery".

**Anonymous Referee #2**

The authors use a new dynamical linear modelling method to identify slowly varying trends in the global ozone profile. They find increasing ozone in the upper stratosphere since the late 1990s, little change in mid-stratospheric ozone since the 1990s, but significantly declining lower stratospheric ozone over the entire 1985 to 2016 period. These results are generally consistent with a number of recent ozone trend studies. However, this study is the first to focus on the lower stratospheric decline, whereas many other studies do not show significant decline in the lower stratosphere, and/or do not focus on this region. The decline in lower stratospheric ozone would explain why, so far, no significant increases in total column ozone have been observed, despite the decline of ozone depleting substances since the late 1990s, and despite expectations from model simulations. Model simulations, in fact, indicate that lower stratospheric ozone should be increasing. As pointed out by the authors, a decline in lower stratospheric ozone, as reported here, seriously questions our understanding of global ozone trends, and our ability to model them.

1 General Comments

Overall, this is a good paper, well suited for ACP, and deserving publication. There are aspects, however, where I am not certain, and where I feel a bit more scepticism would be appropriate.

1. Originators of the merged SBUV (Frith et al., 2017, https://doi.org/10.5194/ acp-2017-412), CCI (Sofieva et al. 2017, https://doi.org/10.5194/acp-2017-598) and SOO (Bourassa et al. 2017, https://doi.org/10.5194/amt-2017-229) data sets do not have the same confidence as the authors into the stability and reliability of their ozone records in the lowermost stratosphere. Frith et al., (2017) do not report trends below 30 hPa (25 km), Sofieva et al. (2017) do not report trends below 20 km (50 to 70 hPa). Bourassa et al. (2017) do not report trends below 18 km (70 hPa). Given this, more caution on the reliability of lower stratospheric ozone (147/100 to 32 hPa; 13/16 to 27 km) in this study would be appropriate. Uncertainties in this region are large, and easily exceed 5% (see e.g. Fig. 9 of Sofieva et al.). With such low accuracy, small trends like the one reported here (-2 DU / decade, for a value of maybe 100 DU), of the order of a few percent per decade, are always questionable, and have to be put into perspective.

We agree that uncertainties remain a significant problem here, not least because actually quantifying the measurement uncertainties themselves remains a difficult task that is unresolved (e.g. Harris et al., 2015). We have added the following (in addition to other points about data uncertainties addressed in other responses to referee comments; see elsewhere here):

- In the introduction: "**It should be noted that absolute uncertainties between limb sounding instruments have been reported to be up to ~10-15% near 16 km (Tegtmeier et al., 2013), which reduces confidence in variability and trends in the lower stratosphere.**"
- In the 'ozone data' section: "These data are monthly, zonally averaged, homogenised, and bias-corrected ozone datasets. **Nevertheless, merged product uncertainties remain large in the lower stratosphere, with estimated monthly uncertainties of 3-9% in SAGE-II-CCI-OMPS (Sofieva et al., 2017), and drifts of ~1% per decade in the OSIRIS period of SAGE-II-OSIRIS-OMPS (Bourassa et al., 2017). Additional uncertainties remain, such as those in the SBUV (vertically resolved) composites due to very low resolution in the lower stratosphere (FrithKramarova2014), and the unquantified uncertainties that result from the conversion from number-density to vmr in the SWOOSH and GOZCARDS composites that require information about local temperature. We note, however, that formal definitions and calculations of uncertainties vary between composites and cannot necessarily be directly compared (Harris et al., 2015; Ball et al., 2017).**"

2. Figure 3 demonstrates steps in the lower stratospheric ozone time series. These might be due to instrumental changes relevant for all merged data sets. The large downward step by about 2 DU in 2004/2005 occurs at exactly the time when the SWOOSH and GOZCARDS merged ozone records switch from the very sparsely sampling solar occultation SAGE II instrument (operating until 2005, with even more reduced sampling since 2001) to the very densely sampling Microwave Limb Sounder (since 2004), an instrument with characteristics very different from SAGE II. Similar things apply for the CCI data set at the switch-over from SAGE II to ENVISAT instruments around 2002/2003, where, e.g., Figs. 8 and 9 from Sofieva et al. (2017) demonstrate the large changes in sampling and the large uncertainties in the lowermost stratosphere. These changes could be very important for time series and for trends in the lowermost stratosphere. I think the authors should add some more caution here. How do the curves look for CCI and SOO?

Starting with the final comment here by the referee, we show here (Fig below) the curves for the lower stratosphere from SAGE-II-CCI-OMPS and SAGE-II-OSIRIS-OMPS, in addition to

that of Merged-SWOOSH/GOZCARDS as in Fig 3d. Note that, as presented in Fig A5, while much of the data (typically greater than 70%) exists in each latitude band, when integrating over 60S-60N, this reduces to 50% or less in the two additional composites, so curves are less well constrained. Even so, the reduced uncertainty from having less data will be included in the posteriors presented in the manuscript. It is clear from the figure below that similar changes are occurring in these two composites in the lower stratosphere, as in the Merged-SWOOSH/GOZCARDS composite.

We agree it is possible that a step or rapid change related to the data itself might have occurred during this time, although given that the overlap/merging occurs over slightly different times in the composites, such a decrease is more likely to be reasonable than if we were only considering one composite; in addition there is the qualitatively agreeing result from the spatially resolved SBUV (Fig 1). As such we have added in at the end of the discussion of global lower stratospheric trends (in bold): "…lower stratospheric ozone has seen a continuous and uninterrupted decrease. **We note that a large proportion of the post-1997 decline occurred between 2003 and 2006, during which overlaps and switch-overs between different combinations of instrument data were used to form the composites, most notably from the low-sampling SAGE-II instrument that ended operation in 2005, although all composites display similar behaviour, and overlaps and switch-overs between different instrument data occur at different times (see Fig. 1 in both Tummon et al., 2015 and Sofieva et al., 2017).**

[Figure]

3. The tropospheric OMI/MLS column in Fig. 4 does not provide an independent piece of information. It just shows that the difference between OMI total column ozone, which should be very similar to SBUV total column in the present study (and have no trend since about 2000), and MLS stratospheric column (essentially the same as SWOOSH and GOZCARDS used in the present study) has a positive trend. Since upper stratospheric is increasing, this just means that lower stratospheric ozone from MLS (and GOZCARDS, SWOOSH) must be decreasing. While this confirms the findings of the authors, it still hinges on the same MLS data, and does not provide an independent piece of information. Independent information about tropospheric ozone trends must come from somewhere else. However, recent studies show no trend for zonal mean tropical tropospheric ozone based on GOME/SCIAMACHY/GOME2 data (Leventidou et al., 2017,

https://doi.org/10.5194/acp-2017-815), or provide little confidence on our ability to identify large scale tropospheric ozone trends (e.g. Cooper et al. 2014, http://doi.org/10.12952/journal.elementa.000029).

The referee raises a fair point regarding independence, although we do explicitly make the statement that OMI/MLS results are not independent from Merged-SWOOSH/GOZCARDS as Aura/MLS forms a part of this composite post-2004. The 2005-2016 period also provides some evidence that strengthens our result – this period is free of the issue of the rapid decline shown between 2002 and 2006. First, it has been shown that GOME-SCHIAMACHY-GOME2, OMI and SBUV are in good agreement in this latter period (Chehade et al., 2014; McPeters et al., 2015; Weber et al., 2017), meaning independent total column ozone observations agree. Second, for this period Merged-SWO./GC., SAGE-II/CCI/OMPS and SAGE-II/OSIRIS/OMPS all show very similar behaviour in the lower stratosphere and are also independent (see figure above), while all show an increase in the upper stratosphere. While not significant for 2005-2016, the no-change or slight decrease in total column ozone together with the increasing ozone in the troposphere from OMI/MLS over 2005-2016 hints at a decrease in stratospheric column ozone. And, since upper stratospheric ozone in other composites is all seen to be rising, this implies a decline in lower stratospheric ozone. On the other hand, if tropospheric ozone is not increasing and upper stratospheric ozone is increasing, then lower stratospheric ozone should be decreasing. We further respond to comments on tropospheric ozone below.

I acknowledge that in parts of the manuscript, the authors are mentioning these open questions. However, I do feel that they should be a more integral part of the manuscript. Therefore, I suggest that the authors reword / change parts of their manuscript, to better reflect these open questions. Below, I'll indicate in more detail which specific parts I am talking about.

2 Detailed Comments

Title: Given all the uncertainties, I would put a question mark behind the title.

We agree that there are a lot of uncertainties, and we can see the reviewer's perspective that the title might come across as being more confident than the additional uncertainties might allow. However, while each component of the puzzle, and the observations, i.e. total column, upper stratospheric, lower stratospheric, and tropospheric ozone each have their own uncertainties and caveats, together they provide re-enforcing evidence of the conclusion we have come to, and the data represents the best estimates we currently have available. Therefore, rather than adding a question mark, we suggest an alternative to make it clear that we simply provide evidence for our conclusion and change the title to: **"Evidence for a continuous decline in lower stratospheric ozone offsetting ozone layer recovery".**

Lines 6, 13, 14, . . . : I find the abbreviations TCO, StCO, TrCO unnecessary and annoying. Every time I read them, I have to re-think what is meant. I would prefer to have them spelled out, throughout the manuscript: total column ozone, stratospheric column ozone, tropospheric column ozone. Text length would not change.

All abbreviations of the TCO, PCO, StCO, and TrCO variety have been written out in their full form.

Lines 14-15: Delete "and harmful to respiratory health". This is irrelevant in the context of the paper. In fact, given the uncertainties mentioned above, I think the entire sentence about tropospheric ozone increase could be omitted, or at least reworded. Certainly, tropospheric ozone changes are not investigated thoroughly in the present paper.

Done for the first part. We have reworded the sentence for the second part to: **"We find that globally, total column ozone appears not to have decreased because of likely increases in tropospheric column ozone that compensate for the stratospheric decreases."**

Lines 17 to 20: Not investigated in the paper. The last sentence should be removed.

Agreed – we have integrated this information into the conclusions (bold text in extracts below):

"Less significant … and a small, additional offset of GHG radiative forcing (RF) **leading to a minor reduction in the warming of the climate** (Randel and Thompson, 2011). Most significantly, restoration of the ozone layer is essential to reducing the harmful effects of solar UV radiation **that impact surface life, and human and ecosystem health** (Slaper et al., 1996)"

**"Further reductions in lower stratospheric ozone may lead to a small reduction in the warming of the climate, and a reduced ozone layer may also permit an increase in harmful ultra-violet (UV) radiation at the surface that would impact human and ecosystem health."**

Line 30: I think a reference is required here.

In hindsight, this sentence is ambiguous – we now revise it to indicate that 1997 is thought to be approximately the time that decreases in ozone ceased and, of course, this could not have been detected until sufficient data had accumulated. Thus we revise this sentence to state:
**"… by the mid-2000s it had become apparent that a decline in total column ozone had stopped at almost all non-polar latitudes since around 1997 (WMO, 2006)."**

Lines 32-33: I think we are far from attribution in the IPCC sense. Therefore I would suggest to delete "an attribution", insert "due" before "to decreasing ODS", and replace "possible" by "reported."

Done.

Line 34: after "rates" add "and by accelerating ozone transport through the meridional Brewer Dobson Circulation".

Done.

Line 36: A reference is needed here.

Added the following references: Revell et al., 2012 "The sensitivity of stratospheric ozone changes through the 21st century to N2O and CH4" (ACP); Nowack et al., 2014 "A large ozone-circulation feedback and its implications for global warming assessments" (Nature Climate Change).

Lines 37 to 97: This is quite longish and wordy, and seems to have been written in several steps and at different times. I would recommend to shorten and compact this: The paragraph about total ozone (around line 40) should include the newest results from Weber et al. (2017, https://doi.org/10.5194/acp-2017-853). The part about differences between MLR, EESC, PWLT (lines 42 to 60) should be moved to the end (line 98), and should be shortened and combined with the paragraph starting in line 98. The paragraph around line 90 should mention more about the general uncertainties of ozone measurements in the lower-most stratosphere, see also my general remark above. Overall, I think the entire introduction could be shortened by 20 to 30%, because many things are clear to an ACP audience, and are also mentioned again later.

Weber et al. 2017 has now been included. Paragraphs around 40 and 98 have been halved in length, merged and partly rewritten. The paragraph around line 90 has more information on uncertainties. The introduction has been made, overall, more concise.

Line 65: Frith et al., 2017, https://doi.org/10.5194/acp-2017-412, should be added here.

Done.

Line 77: This could/ should also include relevant references from lines 64, 65.

Added Harris et al. (2015) and Steinbrecht et al. (2017) here.

Line 84: I think this is a key point here: Instrumental uncertainties are 10 to 15%, and the "observed" lower stratospheric ozone decline is only about 2 DU out of maybe 100 DU. Can we believe a 2% effect measured by a system that is only accurate to within 10 or 15% ?

Moved and changed the last sentence of previous paragraph to just after the first sentence of this paragraph such that it now reads: "**Absolute uncertainties between limb sounding instruments have been reported to be up to ~10-15% near 16 km (Tegtmeier et al., 2013), which should be accounted for from bias corrections when composites are constructed, but which may also reduce confidence in variability and trends in the lower stratosphere.**"

However, absolute uncertainty does not translate to relative uncertainties, so it's not clear the situation is as bad as suggested (though see other additional text on uncertainties added and discussed in the other points in response to reviewer comments). However, such large uncertainties may indicate that, due to different vertical observing kernels, reported layers may not quite match and so that the daisy-chained timeseries may not then really represent the same part of the stratosphere at all times, though this is our conjecture and needs further investigation. As such, we have added the last part of the sentence above: "**which may also reduce confidence in variability and trends in the lower stratosphere.**"

Lines 86-87: There are good reasons, why many of the data providers do not trust derived trends below 18 to 20 km. See e.g. Fig. 9 of Sofieva et al. (2017).

We have addressed this in other comments/additions to the manuscript on data uncertainties (see other responses here).

Line 100: The work by Damadeo et al. (2014, https://doi.org/10.5194/ acp-14-13455-2014; 2017, https://doi.org/10.5194/acp-2017-575) should be referenced as well.

Done.

Line 106, 107: It is no big achievement to not report ozone changes as percentages. Suggest to drop ", i.e. . . . in percentage"

Done.

Line 115: The sentence does not make sense. Something is missing here.

We have reformulated this sentence to read:
**"We also make use of updated ozone composites extended to 2015/6 (section 3) and analyse them with the DLM approach. We begin by considering relative percentage changes since 1998 to put these new data in the context of previously reported relative trends, which are usually reported from 20 km upwards, but here we extend down to the tropopause"**

Line 121: Replace "trends have been" by "about ozone trends"?

Done.

Lines 125 to 140: Reduce duplications with what has already been said in lines 42 to 60.

We feel its important to mention these points in both the introduction and the more specific, detailed sections later and that a slight repetition of some points in the introduction later is useful for readers interested in an overall understanding (without reading the technical sections in detail). Having said that, we have made an effort to address unnecessary repetition such that the introduction has been reordered, partly rewritten, and we have reduced some duplication between these two paragraphs.

Line 139: Probably better to say "PWLT" instead of "linear trend".

Done.

Line 144: What is the correlation between F30 and F10.7 on the time steps used in the present analysis? What is the correlation of the two proxies with ozone, and are there any significant differences between F30 and F10.7 for this type of ozone trend analysis?

We use the 30 cm radio flux (F30) instead of the F10.7 simply because it better represents the UV variability it is meant to represent when solar activity is high (on monthly timescales); it is more/very similar to the Mg-II index. A paper is pending where this is made explicit. It has an effect on the solar cycle estimate below 7 hPa of order of 100% in the mean, though we have not conducted tests on its effect on the trend. It is likely the effect is small, but it remains more representative than F10.7. Essentially, at medium and low activity levels they are very similar, but their relative variability diverges at high activity levels when a non-linear relationship becomes more apparent. We point the referee to Dudok de Wit et al., 2014 for more information.

Line 157: Delete "being considered"?

Done.

Lines 166 to 170: Since a lot of these data sets have changed recently, e.g. from Tummon et al., 2015 to Steinbrecht et al. 2017, I think it is absolutely necessary to indicate already here which data and versions were in fact used. This may require a small table. Mentioning the SPARC LOTUS initiative, which brought together many of the datasets, would also be a good thing.

This has been added, including a new table.

Lines 185, 187: I think Frith et al. (2017) needs to be added here.

Done.

Lines 191 to 199: It would be better to drop this here, and include the relevant information into the paragraph from lines 166 to 177.

Done.

Lines 206 to 213: Given my major comment above, and the general question about relevance / independent information content of the OMI-MLS data set: Maybe drop the entire paragraph? I think a short mention in the description of Fig. 4 would be enough. Only if the authors decide to make a stronger point about tropospheric increases, e.g., by adding an analysis of ozone trends from ozone sounding stations, then a separate sub-section would be appropriate.

See response to major comment above.

Line 239: As mentioned in my major comments, I am still only ≈90% convinced that ozone has declined in the lower stratosphere. Therefore, I suggest to replace "clearly indicate" by "give a strong indication".

We have changed "clearly indicate" to "strongly indicate".

Line 250: I think more words of caution about the high variability of ozone in the lowermost stratosphere, and about the poorer accuracy of the measurements there (compared to the mid- and upper stratosphere) would be required here. See also Fig. 9 of Sofieva et al. 2017.

It is true that higher variability increases uncertainty, but only if the regression model is unable to account for that increased variability. That is indeed the case in the lower stratosphere. However, the result is that the posterior distributions widen to accommodate the reduced ability of the regression model to capture variability and therefore the larger uncertainty, and so the confidence in a decline already accounts for the larger variability in this region. Thus, we would argue that this high variability does not interfere with the confidence we estimate from our posteriors. However, we would agree that the larger variability could affect the accuracy of the data merging through different satellite vertical resolution and sampling. We address this as specified by both referees elsewhere when discussing both the global lower stratospheric trends (end of section 4.2), and the ozone data sets (section 3.1; previously 3.0.1). Additionally, we have changed the last sentence from "... means we are now able to confidently identify changes in the lower stratosphere." to "... **increases our confidence in the identified changes in the lower stratosphere.**"

Around lines 300, 327: Also compare with / better compare to Weber et al. 2017.

Weber et al., 2017 did do a similar comparison to trends as Frith et al., 2017 (~line 300), so we have additionally mentioned it here as requested.

Line 310: Again: This fairly small change by 1.5 DU is challenging the limited accuracy of the instruments, which is around 1% or 3 DU for total column ozone, and around 5 to 10% for the lowermost stratosphere (= 2 to 5 DU, assuming 50 DU sit in the lower stratosphere). A large part of the observed 2 DU drop in the lower stratosphere around 2004/5 hinges on poorly sampled data from SAGE II, at the end of its lifetime.

Agreed – we have added a comment discussing this at the end of section 4.2.

Lines 331 to 366: Given my major comments about the OMI/MLS tropospheric ozone results, and in favor of conciseness of the paper: Would it not be much better to drop much of this discussion, drop Figs. 4 and A13? Instead just mention possible tropospheric ozone increases from OMI/MLS and other, more independent sources of information and put them into perspective. Essentially, this could be done with an expansion of the paragraph in lines 367 to 376. The main messages of the paper would remain. Questionable information would disappear, and conciseness would be improved.

We consider this to be an important part of our work that contributes to our conclusions. We accept that tropospheric ozone from satellite measurements may suffer from significant uncertainty (although current peer-reviewed publications are lacking and the TOAR project is not yet complete), but the missing tropospheric component of the total column is essential to square the difference between the total and stratospheric column. We agree, it is reasonable to question the uncertainties of the stratospheric column, and just considering uncertainties on total and stratosphere alone (without considering the trend analysis) may lead to a conclusion that the diverging trends are simply due to large uncertainties. However, if one does not accept the lower stratospheric trends, but accepts confidence in total and tropospheric ozone, then the conclusion of a decline in stratospheric ozone logically follows.

Having said this, we shortened this part from 'Returning to the OMI/MLS… [to] … is indeed from increasing tropospheric ozone.' We reiterate that we included the following to specifically highlight to the reader that uncertainties and trends in tropospheric ozone still need to be fully understood and quantified: "A deeper investigation is needed to understand difference in the contributions of tropospheric column ozone and stratospheric column ozone to total column ozone, especially considering uncertainties carefully, but this is beyond the scope of this work." and then follow with specific comments that "estimates… suggest a large range of uncertainty".

Lines 418 to 427: Is this paragraph necessary? I think it could easily be dropped. The entire section 4.4 is quite long and wordy. I think it could be shortened and made more concise.

Following this, and the first reviewer's suggestion (to line 441), these paragraphs have been significantly shortened and merged. Please see comments/response to first reviewer for more details.

Line 452: Here is one place, out of many, where TCO left me very confused. I was thinking of TCO = tropospheric column ozone, and saw little sense in the paragraph. As mentioned, spelling out TCO, StCO, TrCO, . . . would help readers like me.

All abbreviations of the TCO, PCO, StCO, and TrCO variety have been written out in their full form.

Line 475: If you do the numbers, this is still a very small effect for past total ozone columns, maybe 0.2 DU per decade. I think this should be said here.

Are you referring to the VSLS influence (e.g. from Fig 4 of Hossaini et al., 2016, Nat Comms)? If so, it is not clear how exactly 0.2 DU per decade were estimated. We might suggest a slightly larger effect, maybe 0.5-1 DU per decade globally, which might be argued as relatively big. Nevertheless the effects are smallest at mid and equatorial latitudes, so the suggestion here is valid. Therefore, rather than quote specific quantities, we simply state that the effect is expected to be small at mid and equatorial latitudes: "**though the effect outside of the polar latitudes is expected to be quite small**".

Lines 458 to 494: Again, I think this is quite long and wordy, and would benefit from substantial shortening.

We have shortened these two paragraphs by nearly 50% without losing the main points.

Figure A7: Can you show similar plots for the altitudes where it really matters, e.g. 18 km? And also include SWOOSH / GOZCARDS?

The aim of this figure is to show that the reason that SOO shows an increase larger than CCI (and SWOOSH/GOZCARDS) is because of a step. At 18km, all three composites shown have the same trend at 18 km. With that in mind, we show the 60-35S band at 17 km where in Fig 1 there is a clear positive trend in this region mainly at 40-30 S in SOO. We see in the updated supplementary plot with this additional altitude that there is an undulation and upward 'drift' between the two number density composites after 1998, starting off again with SOO below CCI, especially evident during the OSIRIS period of SOO and/or the CCI period in CCI. We include the Merged-SWOOSH/GOZCARDS data at 10 hPa for the 30 km NH example, and 83 hPa for the 17 km SH example.

To summarize again: I think this is a good paper. I think it would benefit greatly from addressing my major points raised above. It would also benefit substantially from fleshing out redundancies and shortening the text. When this has been done, I fully recommend publication.

References:

Chehade, W., Weber, M., and Burrows, J. P.: Total ozone trends and variability during 1979-2012 from merged data sets of various satellites, Atmospheric Chemistry & Physics, 14, 7059–7074, doi:10.5194/acp-14-7059-2014, 2014.
Dudok de Wit et al., J. Space Weather Space Clim 4 (2014) A06, 10.1051/swsc/2014003
McPeters, R. D., Frith, S., and Labow, G. J.: OMI total column ozone: extending the long-term data record, Atmospheric Measurement Techniques, 8, 4845–4850, doi:10.5194/amt-8-4845-2015, 2015.

Weber, M., Coldewey-Egbers, M., Fioletov, V. E., Frith, S. M., Wild, J. D., Burrows, J. P., Long, C. S., and Loyola, D.: Total ozone trends from 1979 to 2016 derived from five merged observational datasets – the emergence into ozone recovery, Atmos. Chem. Phys. Discuss., 2017.